# Rethinking Visual Intelligence: Insights from Video Pretraining

**Pablo Acuaviva** [1]  **Aram Davtyan** [1]  **Mariam Hassan** [2]  **Sebastian Stapf** [1]  **Ahmad Rahimi** [2]  **Alexandre Alahi** [2]
**Paolo Favaro** [1]

## Abstract

Large language models (LLMs) have demonstrated that large-scale pretraining enables systems to adapt rapidly to new problems with little supervision in the language domain. This success, however, has not translated as effectively to the visual domain, where models, including LLMs, continue to struggle with compositional understanding, sample efficiency, and general-purpose problem-solving. We investigate Video Diffusion Models (VDMs) as a promising direction for bridging this gap. Pretraining on spatiotemporal data endows these models with strong inductive biases for structure and dynamics, which we hypothesize can support broad task adaptability. To test this, we design a controlled evaluation in which both a pretrained LLM and a pretrained VDM are equipped with lightweight adapters and presented with tasks in their natural modalities. Across benchmarks including ARC-AGI, ConceptARC, visual games, route planning, and cellular automata, VDMs demonstrate higher data efficiency than their language counterparts. Taken together, our results indicate that video pretraining offers inductive biases that support progress toward visual foundation models. Project page:
https://pabloacuaviva.github.io/rethinking-visual-intelligence/

## 1. Introduction

Foundation models have reshaped natural language processing by showing that large-scale pretraining can equip models with broad knowledge and strong inductive priors. This foundation allows models to adapt quickly and effectively to new tasks through techniques like in-context learning (Brown et al., 2020) and parameter-efficient fine-tuning (Liu et al., 2022), achieving strong performance with minimal supervision. The success of Large Language Models (LLMs) illustrates how scale and pretraining can create systems that generalize across diverse problems. Achieving a similar level of versatility in vision, however, remains largely unexplored and a major challenge. Despite recent breakthroughs in image and video generation (Labs, 2025; Polyak et al., 2024; Qin et al., 2024), vision models are not yet on par with LLMs when it comes to compositional skills, sample efficiency, and versatility in problem solving.

Video Diffusion Models (VDMs) represent an exciting direction for narrowing this gap. Pretraining on rich spatiotemporal data endows them with strong inductive biases for spatial structure and temporal dynamics (Blattmann et al., 2023; Google DeepMind, 2025; Wu et al., 2025), which we hypothesize can be harnessed for structured visual understanding. We move beyond treating videos as mere generative artifacts and instead regard them as a natural representation for problem solving, where tasks are expressed as transformations unfolding over time. Building on this perspective, we introduce a simple and general framework for adapting VDMs to a broad class of visual tasks and evaluate them head-to-head with equally adapted LLMs (see Figure 1). This setup allows us to test whether large-scale video pretraining offers a complementary foundation for structured visual problem-solving, contrasting the strengths of visually grounded models with those of symbolically trained language models.

Each task is represented consistently but adapted to each model family's modality: LLMs operate in a text-to-text setting, where inputs and outputs are serialized into structured text, while VDMs receive an image-to-image formulation, where input–output pairs are rendered as short videos to model the task as a temporal transformation. Both model families use identical LoRA-based (Hu et al., 2022) adaptation: adapters are inserted at corresponding layers, pretrained backbones remain frozen, and only lightweight parameters are updated. This symmetry provides a controlled basis for comparison and isolates the impact of video pretraining on structured visual understanding.

Our contributions are as follows:

[1]Computer Vision Group, University of Bern, Switzerland
[2]VITA Lab, EPFL, Lausanne, Switzerland. Correspondence to: Pablo Acuaviva <pablo.acuavivahuertos@unibe.ch>.

*Proceedings of the $43^{rd}$ International Conference on Machine Learning*, Seoul, South Korea. PMLR 306, 2026. Copyright 2026 by the author(s).

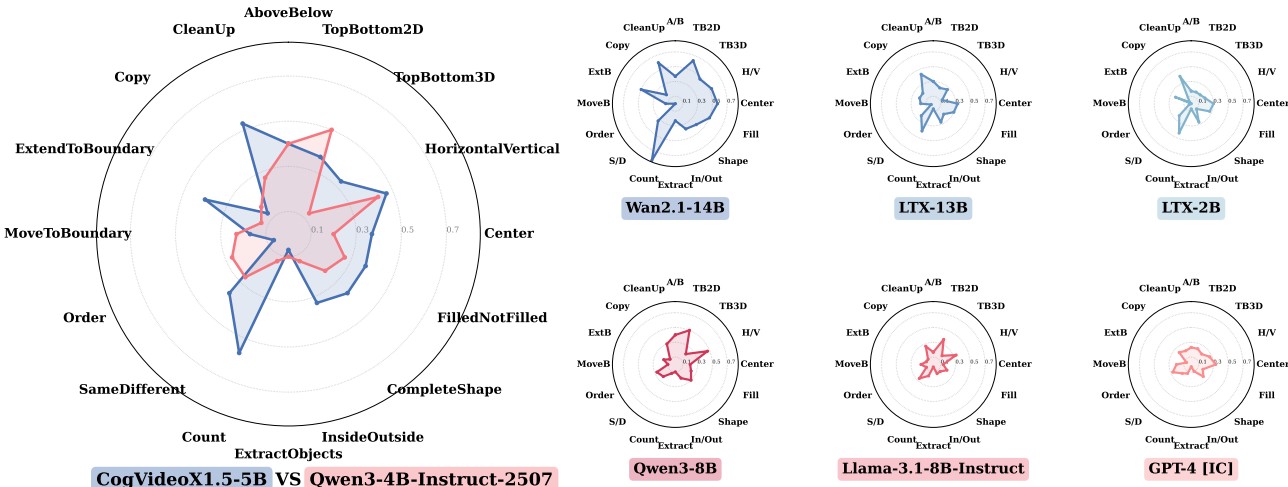

*Figure 1.* ConceptARC competencies between **VDMs** and **LLMs** , GPT-4 [IC] added for additional reference.

1. A unified framework for adapting VDMs to image-to-image visual tasks by reframing examples as temporal sequences.

2. A controlled evaluation setting where both VDMs and LLMs are fine-tuned with LoRA-based adaptation, enabling direct comparison.

3. Empirical evidence that VDMs benefit from video pretraining for visual intelligence, hinting at a path toward flexible visual foundation models with both generative and problem-solving strengths.

## 2. Related Work

**Language Foundation Models.** LLMs have demonstrated remarkable generalization and adaptability to new tasks with minimal supervision, mainly due to their large-scale pretraining on diverse text corpora (Brown et al., 2020; Chowdhery et al., 2023). Their extensive pretraining equips LLMs with rich knowledge and strong inductive biases, enabling them to perform few-shot learning (Brown et al., 2020) and in-context learning (Coda-Forno et al., 2023), where models learn new tasks only by observing a handful of examples. Parameter-efficient finetuning methods like LoRA (Hu et al., 2022) extend this adaptability allowing LLMs to specialize to new domains while the backbone is completely frozen (Liao et al., 2025). Together, these capabilities make LLMs highly flexible and scalable problem solvers. In this paper, we leverage this adaptability to compare the data efficiency of LLMs and VDMs across diverse visual tasks.

**Video Diffusion Models.** Diffusion-based generative models have recently achieved remarkable progress in video synthesis. Pioneering approaches such as CogVideo (Hong et al., 2022) and (Villegas et al., 2022) introduced scalable

architectures for text-to-video generation. More recent models like Sora (Qin et al., 2024), MovieGen (Polyak et al., 2024), Veo 3 (Google DeepMind, 2025), and CogVideoX (Yang et al., 2024) set new standards for quality and realism. Recent work has investigated controllable video generation (NVIDIA et al., 2025; Hassan et al., 2025; Kanervisto et al., 2025), with the goal of producing realistic, high-quality videos while allowing precise control over motion and dynamics. These methods emphasize modeling dynamic environments and predicting plausible future states conditioned on past observations and control inputs.

**Visual Foundation Models** Recent work has investigated the use of generative models as generalist vision models. Methods such as image inpainting for visual prompting (Bar et al., 2022) and image-based in-context learning (Wang et al., 2023a) demonstrate that structured inputs can enable these models to solve diverse tasks. Diffusion models have further been extended to in-context learning (Wang et al., 2023b), instruction following across heterogeneous tasks (Geng et al., 2024), and broader computer vision problem solving (Zhao et al., 2025). Sequential modeling has been proposed as a unified interface for scaling vision models (Bai et al., 2024). Building on this line of work, (Lin et al., 2025) train CogVideoX1.5 with temporal in-context prompts for multi-task learning, but their focus remains on broad computer vision benchmarks rather than visual intelligence, and their method requires extensive training[1].

Our approach does not attempt to build a foundation model from scratch. Instead, we investigate whether a pretrained VDM, pretrained extensively on next-frame prediction, can begin to exhibit the properties expected of visual founda-

---

[1]We add qualitative results on standard computer vision tasks in the Appendix to show that our framework can also be extended to this setting.

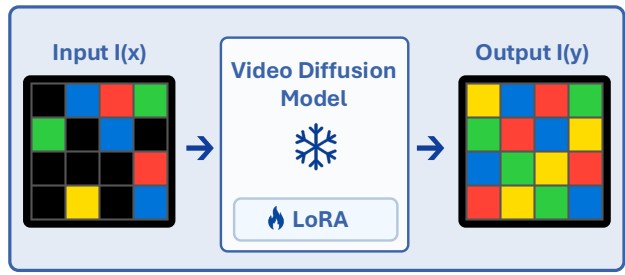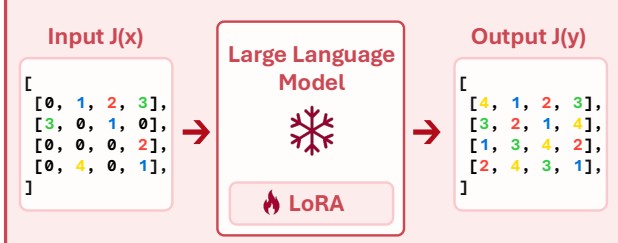

*Figure 2.* Simplified diagram of comparison protocol between VDMs and LLMs.

tion models by leveraging inductive biases gained through spatiotemporal pretraining.

## 3. Methodology

### 3.1. Setup and Comparison Protocol

We adopt the definition of intelligence proposed by (Chollet, 2019):

> *The intelligence of a system is a measure of its skill acquisition efficiency over a scope of tasks with respect to priors, experience, and generalization difficulty.*

This perspective motivates our evaluation design. We focus not only on absolute accuracy but also on how quickly models acquire new capabilities when exposed to limited supervision.

To evaluate our hypothesis, we curate a diverse benchmark of visually grounded tasks that can be specified textually as grid-based problems, including ARC-AGI, Sudoku solving, and route planning. We now describe the evaluation setup in detail.

Let $\mathcal{T}$ denote a task with dataset $\mathcal{D}_{\mathcal{T}} = \{(x_i, y_i)\}_{i=1}^{n}$, where each $x_i$ and $y_i$ is an input-output pair. Each sample is expressed in two complementary modalities:

**Image** An **image pair** $(I(x_i), I(y_i))$, where $I(\cdot)$ deterministically renders RGB images of size $(3 \times H \times W)$.

**Text** A **JSON pair** $(J(x_i), J(y_i))$, where $J(\cdot)$ maps a grid to a compact JSON string.

We serialize samples in a neutral format that avoids domain-specific priors, requiring both models to infer task rules directly from raw representations. Training and evaluation splits are identical across all models to ensure a fair and controlled comparison. VDMs are trained directly on the

image modality using our approach, which we detail in the next section, while LLMs are trained on the text modality. This equivalent comparison is depicted in Figure 2.

We define accuracy as the proportion of test instances where the predicted output *exactly matches* the ground truth grid. For tasks where multiple valid solutions may exist (e.g., *Sudoku*, *Sudoku Mini*, *Hitori*), we filter datasets to ensure each instance has an unique solution. When unique solutions cannot easily be guaranteed, as in *Shortest Path*, we introduce complementary metrics to better capture solution quality (see Section 4.2.2).

To evaluate efficiency of skill acquisition, we consider two complementary settings.

**ARC Family.** We evaluate models on ARC-AGI and ConceptARC, where the challenge is to solve diverse tasks from only 2–5 demonstrations. Following prior work (Moskvichev et al., 2023; Chollet, 2019; Li et al., 2025), we measure how many tasks each model can solve under this minimal supervision regime.

**Structured Visual Tasks.** For these we systematically vary $n$, the number of training examples per task, to trace curves and quantify the rate of skill acquisition rather than focusing solely on endpoint accuracy.

### 3.2. Adapting Video Diffusion Models for I2I

We adapt pretrained VDMs to image-to-image (I2I) prediction tasks by re-framing each input–output pair $(I_{x_i}, I_{y_i})$ as a short *transition video*. This leverages the generative prior of VDMs, while requiring minimal supervision.

**Transition video construction** Each pair $(x_i, y_i)$ is converted into a temporal sequence $v_i = [v_{i,1}, \ldots, v_{i,F}]$, where

$$v_{i,1} = I(x_i), \quad v_{i,F} = I(y_i).$$

Intermediate frames are generated with an interpolation function $\phi$. For example, a *convex interpolation* produces a smooth transition

$$v_{i,f} = (1 - \alpha)\, I(x_i) + \alpha I(y_i),$$

where $\alpha = \frac{f-1}{F-1}$, and $f = 1, \ldots, F$. While a *discrete interpolation* simply holds the input frame for the first half of the sequence and afterwards switches to the output frame:

$$v_{i,f} = \begin{cases} I(x_i), & f \leq F/2, \\ I(y_i), & f > F/2. \end{cases}$$

This yields a dataset $\mathcal{V}_{\mathcal{T}}$ of input-conditioned video trajectories. For our comparisons, we adopt the *discrete interpolation* to avoid introducing any biases.

**Fine-tuning**    We adapt a pretrained VDM by conditioning on the first frame $v_1^0$ and a neutral fixed text embedding $e_{\text{text}}$. Given a noisy video $v^t$ at step $t$, the model predicts noise $\epsilon_\theta$ via

$$\mathcal{L}_{\text{VDM}} = \mathbb{E}_{v^0 \sim \mathcal{V}_{\mathcal{T}}, \epsilon \sim \mathcal{N}(0, \mathbf{I}), t} \left[ \| \epsilon - \epsilon_\theta(v^t, t, c) \|_2^2 \right],$$

with $c = \{v_1^0, e_{\text{text}}\}$. We use LoRA modules for fine-tuning, updating only these additional weights while keeping the pretrained model frozen.

**Inference**    At test time, the model generates predictions through reverse diffusion. The procedure is detailed in Algorithm 1.

---

**Algorithm 1** Inference for **VDM**

---

1: Encode input: $c_{\text{test}} \leftarrow \{I(x_{\text{test}}), e_{\text{text}}\}$
2: Initialize noise: sample $v^T \sim \mathcal{N}(0, \mathbf{I})$
3: Reverse diffusion: recover $v^0$ conditioned on $c_{\text{test}}$
4: Output prediction: $\hat{y} \leftarrow v_F^0$ (final frame)

---

This procedure reframes image-to-image prediction as a conditional video generation problem, enabling efficient adaptation of pretrained VDMs to new tasks.

### 3.3. Adapting Large Language Models

We adapt pretrained LLMs to structured prediction tasks by framing each example as a JSON-to-JSON translation problem.

**Fine-tuning**    We adapt pretrained LLMs using a standard sequence-to-sequence objective. Given tokenized input-output pairs, the model is trained to maximize the likelihood of the target sequence under teacher forcing:

$$\mathcal{L}_{\text{LLM}} = \frac{1}{n} \sum_{i=1}^{n} \sum_{t=1}^{|\mathbf{v}_i|} - \log p_\theta(v_{i,t} \mid \mathbf{u}_i, \mathbf{v}_i^{<t}).$$

We insert LoRA modules into the pretrained backbone, fine-tuning only these lightweight adapters while keeping the majority of parameters frozen.

**Inference**    At test time, predictions are generated autoregressively. The procedure is summarized in Algorithm 2.

---

**Algorithm 2** Inference for **LLM**

---

1: Encode input: $J(x_{\text{test}})$ as JSON string
2: Tokenize and feed sequence into model
3: Autoregressively decode output until termination
4: Return prediction: $\hat{y}$ as JSON string

---

## 4. Experiments

### 4.1. ARC Family

The ARC-AGI benchmark (Chollet, 2019) evaluates an agent's ability to infer and apply abstract patterns through compositional understanding, few-shot learning, and inductive generalization. Each ARC task provides only a handful of input–output examples (typically 2–5), requiring the model to discover the underlying transformation rule and apply it to novel test inputs. This benchmark is widely regarded as a challenging measure of progress in abstraction and generalization.

We follow the evaluation protocol of (Chollet et al., 2024), which allows up to two attempts per test input and counts a question as solved only if all predictions match the ground truth. Quantitative results appear in Table 1, with qualitative examples in Figure 3. For comparison, we also report single-attempt results of commercial LLMs from (Chollet et al., 2024).

*Table 1.* ARC-AGI test performance following the official evaluation protocol (Chollet et al., 2024). Single-attempt results are reported for comparability with commercial LLMs, which are only available under this setting.

| Model | Accuracy (%) |
|---|---|
| **Two-attempts setting** | |
| CogVideoX1.5-5B | 16.75 |
| Qwen3-4B-Instruct-2507 | 8.00 |
| **Single-attempt setting** | |
| CogVideoX1.5-5B | 12.50 |
| Qwen3-4B-Instruct-2507 | 6.75 |
| OpenAI o1-preview | 21.00 |
| Anthropic Claude 3.5 Sonnet | 21.00 |
| OpenAI GPT-4o | 9.00 |
| Google Gemini 1.5 | 8.00 |

We evaluate models on ConceptARC (Moskvichev et al., 2023), a curated variant of ARC designed to systematically

measure visual concept understanding and generalization. ConceptARC groups tasks into 16 concept categories (for example, Above and Below, Center, Count), with each category containing 10 tasks. Each task includes 3 distinct test inputs, creating controlled variation in visual patterns and object relationships while maintaining internal consistency within each concept group. Following the protocol of (Moskvichev et al., 2023), we allow three attempts per test input and mark an input as solved if any attempt is correct. Performance is reported in Figure 1, where we further include as VDMs: Wan2.1-14B (Wang et al., 2025), LTX-13B, LTX-2B (HaCohen et al., 2025), CogVideoX1.5-5B (Yang et al., 2024) and as LLMs: Qwen3-4B-Instruct-2507, Qwen3-8B (Qwen3-4B-Instruct-2507 Team, 2025), Llama3.1-8B (Meta-AI, 2024), and GPT-4 in an IC setting (Moskvichev et al., 2023). Full table with results is included in the Appendix.

These results highlight the importance of strong visual priors: by leveraging representations that capture spatial structure, compositionality, and low-level visual cues, the VDM is able to approach these abstract tasks in a way that improves upon traditional text-centric approaches.

## 4.2. Structured Visual Tasks

From this point onward, we focus on one representative model from each family: **CogVideoX1.5-5B** (Yang et al., 2024) for video diffusion models and **Qwen3-4B-Instruct-2507** (Qwen3-4B-Instruct-2507 Team, 2025) for language models. This pairing aligns model scale while contrasting pretraining modalities, allowing us to examine how different priors influence adaptability to visually grounded tasks.

### 4.2.1. Visual Games

As part of our broader evaluation, we examine performance on a diverse set of five visual games that span both puzzle-solving and board play. These tasks provide an additional perspective on how the models handle structured visual inputs and varying interaction styles. The puzzle-based tasks, *Hitori 5x5* and two versions of *Sudoku* (standard one and *Mini*), focus on solving constraint-based problems in structured grids, where success depends on extracting spatial patterns and enforcing global consistency from local information. The board games, *Connect 4* and *Chess Mate-in-1*, shift attention to game scenarios where the goal is to identify the winning move in a given configuration. Together, these games cover a range of visual layouts and structured objectives, complementing the other tasks explored in this study.

Figure 4 illustrates the trajectory of model performance relative to the volume of training samples, highlighting a distinct trend in scalability. The CogVideoX1.5-5B model exhibits robust scaling capabilities across the majority of the evaluated tasks, effectively outperforming the Qwen3-4B-Instruct-2507 baseline in four out of the five games. This performance gap is most pronounced in logic puzzles like *Sudoku* and *Hitori*, where success depends heavily on the model's ability to parse complex grid structures and maintain spatial consistency. These results suggest that the VDM is better suited for tasks requiring the interpretation of intricate visual compositions and spatial logic, an area where text-optimized models often struggle to generalize solely from linguistic patterns.

Conversely, the chess task presents a notable deviation from this trend, standing as the sole category where Qwen3-4B-Instruct-2507 maintains a performance edge. This exception likely stems from the vast availability of chess notation and game commentary present in textual pretraining corpora, allowing the LLM to internalize rules and strategies (Kuo et al., 2023).

### 4.2.2. Route Planning

To assess the capabilities of our approach for route planning, we conduct comprehensive evaluations across two distinct tasks situated in two-dimensional grid environments: the *Maze* task and the *Shortest Path* task.

In the *Maze* challenge, the primary requirement is for the model to successfully navigate a path originating from the start point at the top-left corner of the grid and terminating at the goal in the bottom-right corner. Conversely, the *Shortest Path* task presents a specific optimization objective: the model is tasked with establishing a connection between two randomly selected arbitrary points, aiming to generate the shortest route possible between them.

To provide a robust analysis of model performance for the *Shortest Path* task, we utilize and report on two complementary metrics:

**Path Success Rate (PSR)** This metric quantifies the reliability of the model by calculating the percentage of total evaluation examples in which the predicted path successfully establishes a continuous, unbroken connection between the designated source and target locations.

**Relative Path Length (RPL)** The second metric is computed exclusively for instances **where a valid path is successfully produced**. It serves as a measure of efficiency and is defined mathematically as:

$$\text{RPL} = \frac{\text{Predicted Path Length}}{\text{Ground-Truth Shortest Path Length}}.$$

It is important to note a nuance in interpreting this metric:

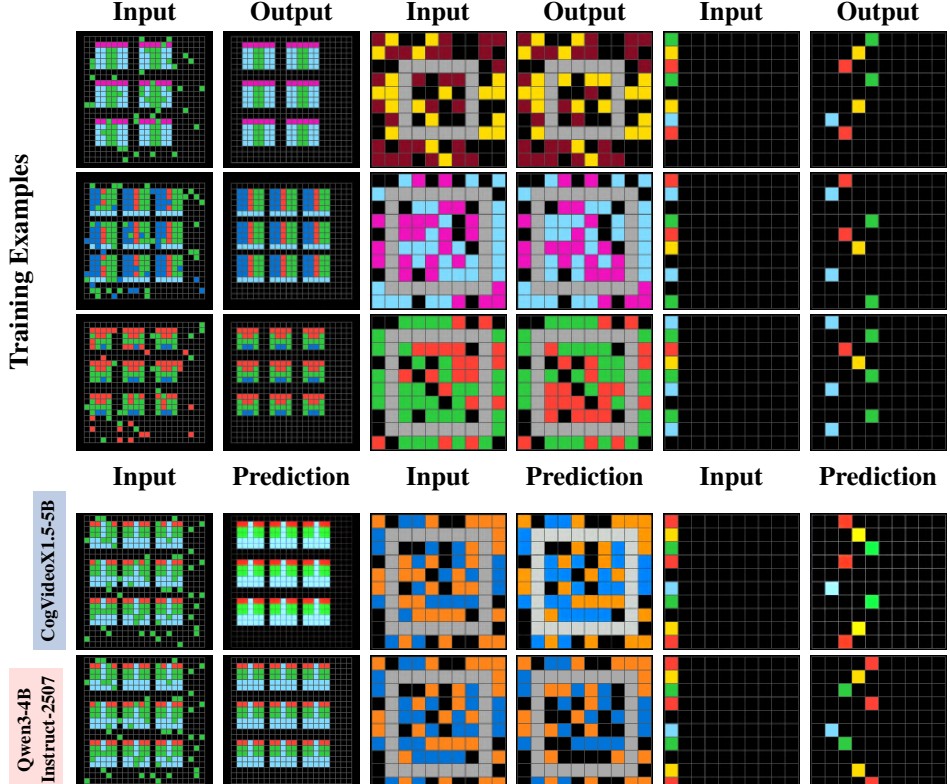

*Figure 3.* Qualitative results on ARC-AGI for problems *0607ce86*, *7ee1c6ea*, and *f45f5ca7*.

the RPL value may increase even when the overall performance of the model improves. This phenomenon occurs because better models are often capable of predicting valid paths for significantly more challenging test cases, which will require longer paths, whereas weaker models might fail these difficult cases entirely. Consequently, a more capable model may construct longer, yet valid, paths for complex scenarios that a baseline model would miss.

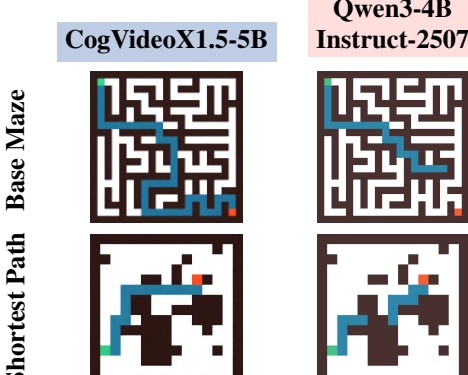

*Figure 6.* Qualitative examples for *Base Maze* and *Shortest Path* tasks, after fine-tuning with $n = 300$ samples.

For *Maze*, we evaluate in two settings: a **matched-scale**

(*Base Maze*) scenario, where both training and evaluation are conducted on $21 \times 21$ mazes to study performance as a function of training sample size; and a **generalization** scenario, where models are trained on smaller $13 \times 13$ grids and tested on larger $21 \times 21$ grids to assess cross-scale generalization (*Maze Generalization*).

Accuracy results are shown in Figure 5. For *Shortest Path*, additional metrics are reported in Table 2. The VDM consistently constructs valid paths with far fewer supervised examples, achieving up to a tenfold reduction in data requirements in low-sample regimes, which underscores its stronger inductive biases relative to the LLM. Moreover, it demonstrates the ability to generalize much quicker from limited training on smaller mazes to larger, more complex ones.

### 4.2.3. CELLULAR AUTOMATA

We evaluate the capacity of both models to capture complex spatial patterns in cellular automata (CA). Our study spans one-dimensional Elementary Cellular Automata (ECA) (Wolfram, 1984), a foundational class of binary-state systems, as well as two-dimensional Life-like Cellular Automata, including Conway's Game of Life (Gardner, 1970), defined by various birth and survival (B/S) rules. Additionally, we consider Langton's ant (Langton, 1986), a deter-

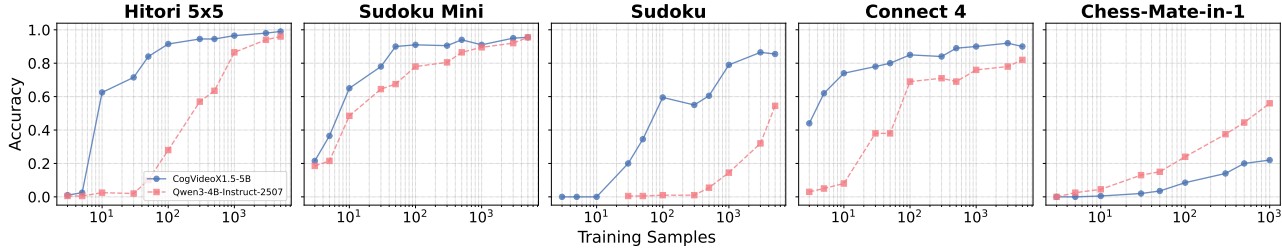

*Figure 4.* Accuracy as a function of training set size for **CogVideoX1.5-5B** and **Qwen3-4B-Instruct-2507** on five visual games.

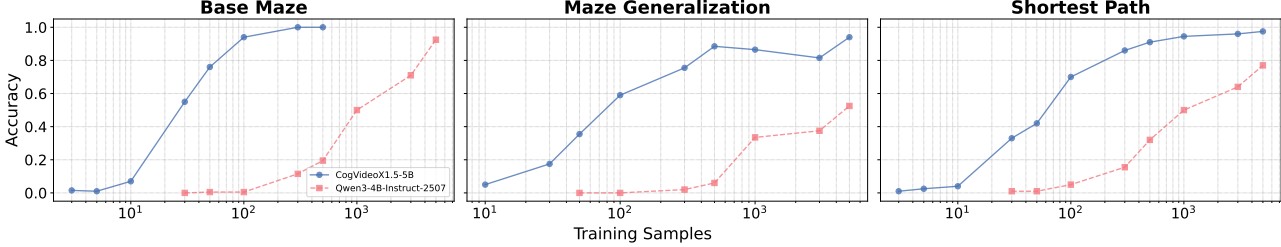

*Figure 5.* Accuracy as a function of training set size for **CogVideoX1.5-5B** and **Qwen3-4B-Instruct-2507** on *Base Maze*, *Maze Generalization*, and *Shortest Path*.

ministic agent-based system, where the task is to predict the complete grid state after $n$ steps of evolution.

In our evaluation of 1D Elementary Cellular Automata (ECA), we selected four representative rules spanning the full range of Wolfram's complexity classes. To strictly define task mastery, we measure the number of training steps required to achieve a validation accuracy exceeding a fixed threshold of $\delta = 0.9$. As illustrated in Figure 7, we analyze the sample efficiency of both architectures across these varying complexity levels. The results indicate a striking parity between the two settings in one-dimensional settings. This suggests that for simpler cellular automata, the generative capabilities of large language models are sufficient to capture the underlying state transitions without requiring domain-specific visual training. We hypothesize that this performance alignment is directly attributable to the task's intrinsic dimensionality. Because 1D ECA dynamics can be effectively represented as linear sequences without a significant loss of topological information, they do not strictly necessitate the complex 2D spatial priors inherent to diffusion architectures. Consequently, the LLM is able to leverage its sequential processing capabilities to model the system dynamics as effectively as the VDM. In this specific context, the VDM's specialized visual inductive biases offer diminishing returns, rendering them less critical than they are in higher-dimensional tasks.

In two-dimensional settings, however, clearer differences emerge (see Figures 9, 10). For Life-like cellular automata, the VDM reaches threshold accuracy with far fewer examples, demonstrating superior data efficiency when spatial locality is paramount. A similar advantage is observed in Langton's ant, where the gap grows larger as the number of steps to be predicted increases. This indicates that the VDM scales more effectively on tasks that demand long-range spatial planning and the maintenance of complex 2D grid states over time.

## 5. Limitations

Our study primarily focuses on grid-based benchmarks, including ARC-AGI, ConceptARC, and synthetic puzzles. This controlled setting provides a systematic way to compare visual and language models, offering a structured interface through which LLMs can express visual understanding. Although these benchmarks do not capture the full diversity of real-world tasks, they are well suited to highlighting the role of modality-aligned pretraining in visual intelligence. Future work should examine whether these insights generalize to more naturalistic and embodied visual environments.

Moreover, our reliance on continuous latent representations introduces a precision bottleneck that is absent from symbolic models. Because VDMs generate continuous RGB values that must be discretized back into grid states, there is a theoretical risk of blurring or loss of high-frequency detail in extremely large or complex grids. This issue extends beyond the current experimental tasks and reflects a broader tradeoff in mechanisms that encode one modality through another rather than operating directly on the target modality. Exploring RGB encoding strategies for different modalities for a variety of tasks is an interesting direction for future

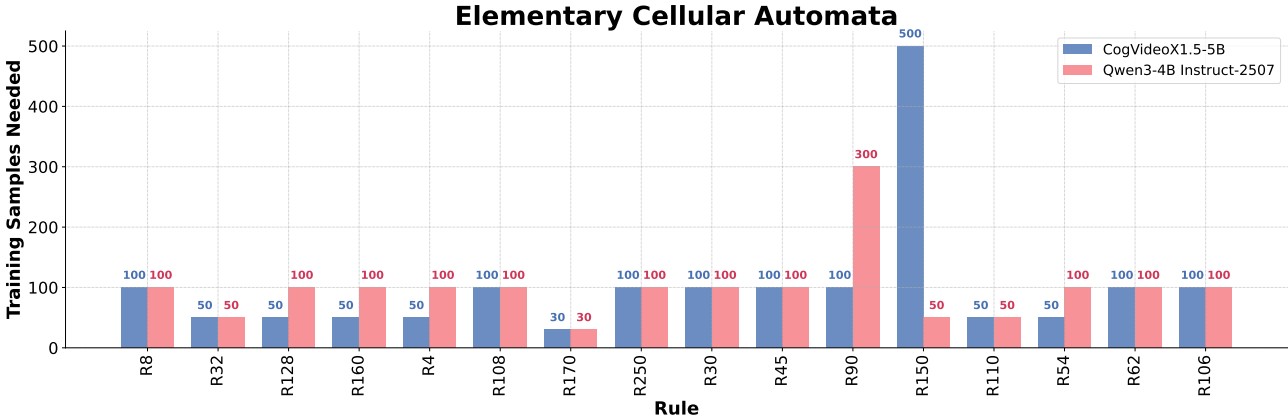

*Figure 7.* Number of training examples required to achieve $\delta \geq 0.9$ accuracy for selected 1D ECA rules (lower is better).

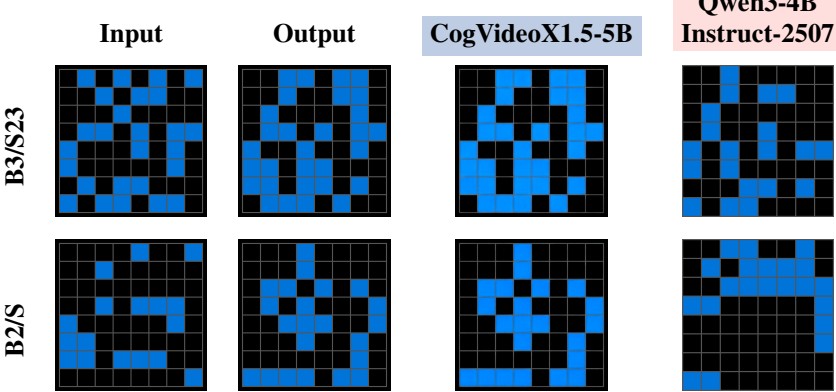

*Figure 8.* Qualitative examples for Life-like cellular automata with rules *B3/S23* and *B2/S* tasks, after fine-tuning with $n = 30$ samples.

*Table 2.* Relative Path Length (RPL) and Path Success Rate (PSR) for both models across training sample sizes for *Shortest Path*.

| Samples | CogVideoX1.5-5B | | Qwen3-4B-Instruct-2507 | |
|---|---|---|---|---|
| | RPL ↓ | PSR ↑ | RPL ↓ | PSR ↑ |
| 3 | 1.005 | 0.115 | – | – |
| 5 | 1.089 | 0.160 | – | – |
| 10 | 1.060 | 0.245 | – | – |
| 30 | 1.028 | 0.670 | 1.020 | 0.015 |
| 50 | 1.013 | 0.645 | 1.038 | 0.060 |
| 100 | 1.017 | 0.870 | 1.025 | 0.205 |
| 300 | 1.007 | 0.940 | 1.040 | 0.530 |
| 500 | 1.005 | 0.985 | 1.019 | 0.605 |
| 1000 | 1.005 | 0.990 | 1.043 | 0.710 |
| 3000 | 1.000 | 0.990 | 1.026 | 0.795 |
| 5000 | 1.001 | 1.000 | 1.016 | 0.870 |

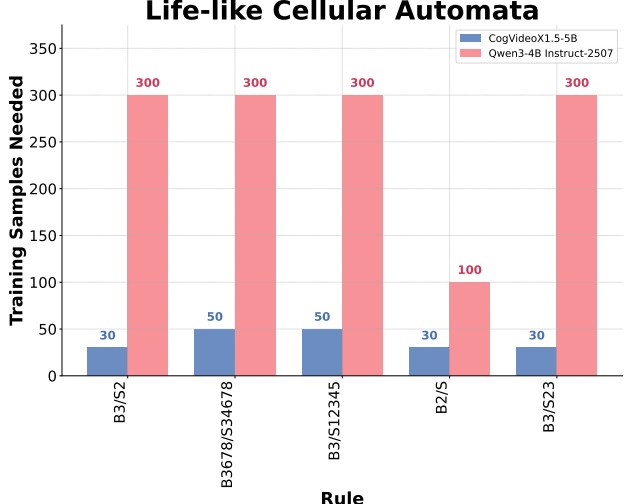

*Figure 9.* Number of training examples required to achieve $\delta \geq 0.9$ accuracy for selected Life-like cellular automata rules (lower is better).

research.

# 6. Conclusions

Our study demonstrates that Video Diffusion Models (VDMs), when adapted via our proposed image-to-image

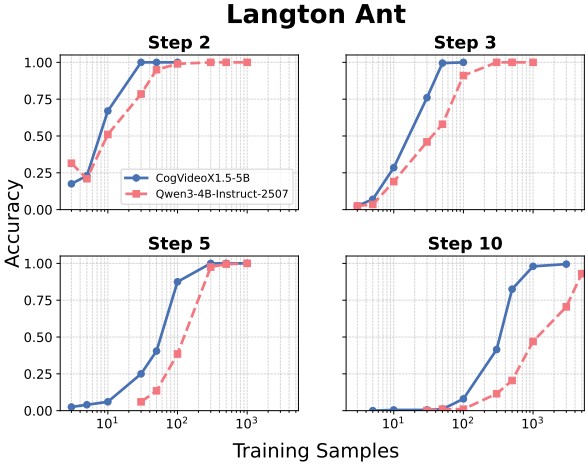

*Figure 10.* Accuracy as a function of training set size for **CogVideoX1.5-5B** and **Qwen3-4B-Instruct-2507** on *Langton's Ant* with a prediction horizon of 2,3,5 and 10.

framework, exhibit superior data efficiency compared to Large Language Models (LLMs) on structured visual tasks. By systematically evaluating performance across benchmarks like ARC-AGI, visual games, and route planning, we observed that the spatiotemporal priors acquired during large-scale video pretraining translate directly into strong inductive biases for visual problem-solving. While LLMs remain competitive in domains rich in symbolic representation or linear sequences, such as Chess or 1D Elementary Cellular Automata, they struggle to match the sample efficiency of VDMs in tasks requiring complex 2D spatial understanding and dynamic simulation. This distinction is particularly evident in navigation and Life-like cellular automata, where VDMs successfully model long-range dependencies and spatial transformations with significantly less supervision.

These results offer insights into the mechanisms of visual learning, presenting distinct implications for both theoretical development and applied deployment. For researchers, the observed contrast in performance on spatially grounded tasks suggests that relying solely on language-based models for visual problems may be fundamentally insufficient; consequently, developing pretraining pipelines that explicitly align representation with the modality's spatiotemporal structure is essential for unlocking rapid skill acquisition. On the practical side, the VDM's robust ability to generalize to new environments indicates its utility in downstream applications such as planning, simulation, and robotics, where a grounded understanding of physical dynamics is paramount. Ultimately, our work validates that video generation models are not merely creative tools but possess latent *visual intelligence*, paving the way for hybrid systems that effectively integrate generative world modeling with robust problem-solving skills.

## Acknowledgements

This work was supported as part of the Swiss AI Initiative by a grant from the Swiss National Supercomputing Centre (CSCS) under project ID a03 on Alps. Pablo Acuaviva, Aram Davtyan and Sebastian Stapf were supported by SNSF Grant 10001278.

Some of the calculations were performed on UBELIX (https://www.id.unibe.ch/hpc), the HPC cluster at the University of Bern.

## Impact Statement

This paper presents work whose goal is to advance the field of Machine Learning. There are many potential societal consequences of our work, none which we feel must be specifically highlighted here.

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

# Appendix

## A. Experimental Details

We report here the detailed computational costs and hyperparameter settings used in our experiments. Tables 3 and 4 summarize the GPU hours required across different tasks, while Tables 5 and 6 provide the LoRA fine-tuning configurations for both VDMs and LLMs.

*Table 3.* GPU hours required for ConceptARC across VDMs and LLMs. Reported hours are wall-clock time and depend on hardware.

| VDM Model (GPU) | Hours | LLM Model (GPU) | Hours |
|---|---|---|---|
| Wan2.1-14B (H100) | 100 | Llama3.1-8B (H100) | 80 |
| LTX-13B (H100) | 95 | Qwen3-8B (2×RTX4090) | 100 |
| CogVideoX1.5-5B (RTX4090) | 130 | Qwen3-4B-Instruct-2507 (RTX4090) | 135 |
| LTX-2B (H100) | 40 | | |

*Table 4.* GPU hours required for ARC-AGI and Structured Visual Tasks. Reported hours are wall-clock time and depend on hardware.

| ARCAGI Model (GPU) | Hours | Structured Task Model (GPU) | Hours |
|---|---|---|---|
| CogVideoX1.5-5B (RTX4090) | 450 | CogVideoX1.5-5B (RTX4090) | 1650 |
| Qwen3-4B-Instruct-2507 (RTX4090) | 475 | Qwen3-4B-Instruct-2507 (RTX4090) | 2000 |

To ensure reproducibility, we also include the fine-tuning hyperparameters for each model. The following two tables detail the LoRA, training, and optimizer configurations used for VDMs (Table 5) and LLMs (Table 6).

*Table 5.* LoRA finetuning configuration for VDM experiments.

| Parameter | LTX-13B | LTX-2B | CogVideoX1.5-5B | Wan2.1-14B |
|---|---|---|---|---|
| *LoRA Configuration* | | | | |
| Rank | 64 | 64 | 64 | 64 |
| Alpha | 64 | 64 | 32 | 32 |
| Target modules | to_q, to_k, to_v, to_out.0, ff.net.0.proj, ff.net.2 | to_q, to_k, to_v, to_out.0, ff.net.0.proj, ff.net.2 | QKVO | – |
| *Training Configuration* | | | | |
| Seed | 42 | 42 | 42 | 42 |
| Batch size | 2 | 4 | 2 | 1 |
| Gradient accumulation steps | 2 | 1 | 1 | 1 |
| *Optimizer Configuration* | | | | |
| Optimizer | AdamW | AdamW | AdamW | AdamW |
| Learning rate | 2e-4 | 2e-4 | 1e-4 | 1e-4 |
| Scheduler | Linear | Linear | Constant | Constant |
| Max grad norm | 1.0 | 1.0 | 1.0 | 0.05 |

*Note.* LoRA ranks differ slightly across model families (VDMs use rank 64, whereas LLMs use rank 32). We verified that performance is largely insensitive to this setting: Qwen3 models with rank 64 perform comparably to rank 32, and CogVideoX1.5-5B models with rank 32 match the reported rank 64 results. In both cases, we report the configuration that yielded stronger results in our initial trials. All reported results in the paper correspond to the configurations shown in the tables.

*Table 6.* LoRA finetuning configuration for LLMs used.

| Parameter | Qwen3-4B-Instruct-2507 | Qwen3-8B | LLaMA-3.1-8B |
|---|---|---|---|
| *LoRA Configuration* | | | |
| Rank | 32 | 32 | 32 |
| Alpha | 32 | 32 | 64 |
| Dropout | 0 | 0 | 0.05 |
| Target modules | q_proj, k_proj, v_proj, o_proj, gate_proj, up_proj, down_proj | q_proj, k_proj, v_proj, o_proj, gate_proj, up_proj, down_proj | q_proj, k_proj, v_proj, o_proj, gate_proj, up_proj, down_proj, lm_head |
| *Model Setup* | | | |
| Max sequence length | 8192 | 8192 | 4096 |
| Random seed | 3407 | 3407 | 3407 |
| *Training Configuration* | | | |
| Batch size per device | 2 | 1 | 1 |
| Effective batch size | 8 | 8 | 8 |
| Gradient accumulation steps | 4 | 8 | 8 |
| Learning rate | 2e-4 | 2e-4 | 2e-4 |
| Scheduler | Linear | Linear | Linear |
| Warmup steps | 5 | 5 | 5 |
| Weight decay | 0.01 | 0.01 | 0.01 |
| *Generation Configuration* | | | |
| Max new tokens | 4096 | 4096 | 4096 |
| Temperature | 0.7 | 0.7 | 0.7 |
| Top-$p$ | 0.8 | 0.8 | 0.8 |
| Top-$k$ | 20 | 20 | 20 |

## B. Task Details

For completeness, we provide additional explanations of the tasks considered in our evaluation. Each subsection introduces a task family and highlights the key rules and objectives, we further provide examples on how the task is encoded into image and text.

### B.1. Visual Games

#### B.1.1. HITORI 5X5

**Objective:** Eliminate cells so that each number appears at most once per row and column.

**Rules:**

1. A number must not be repeated in any row or column.

2. Shaded cells cannot be orthogonally adjacent.

3. All unshaded cells must form a single connected component.

We add an example of the task in Figure 7.

#### B.1.2. SUDOKU

**Objective:** Fill the grid so that all constraints are satisfied.

**Rules:**

1. Each row must contain all required digits without repetition.

2. Each column must contain all required digits without repetition.

3. Each subgrid must contain all required digits without repetition.

**Input**          **Output**

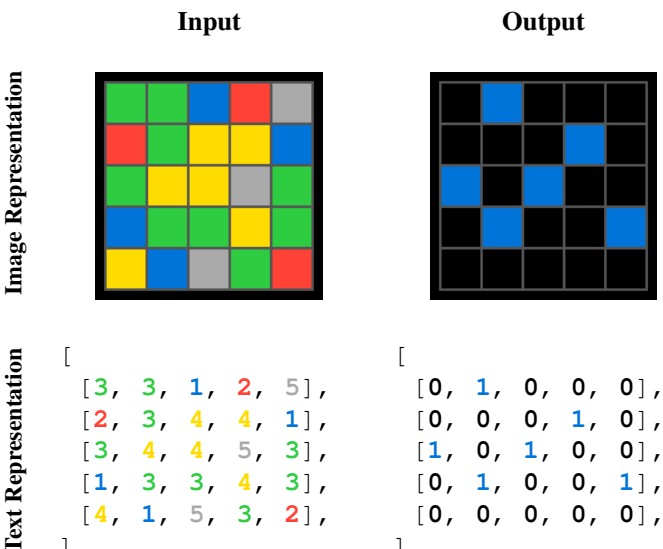

*Table 7.* Example input-output pair for task *Hitori*.

We evaluate two variants: *Mini Sudoku* (4x4 with 2x2 subgrids, see Figure 11) and *Sudoku* (9x9 with 3x3 subgrids, see Figure 12).

### B.1.3. CONNECT 4

**Objective:** Place tokens to align four in a row.

**Rules:**

1. Players alternate dropping tokens into one of the seven columns.

2. A token occupies the lowest available cell in the chosen column.

3. A player wins by forming a horizontal, vertical, or diagonal line of four tokens.

We restrict evaluation to single-move winning scenarios, see Figure 13.

### B.1.4. CHESS MATE-IN-1

**Objective:** Deliver checkmate in a single move. **Rules:**

1. All standard chess movement rules apply.

2. A move is correct only if it results in an immediate checkmate of the opposing king.

To ensure the task is well defined, we filter scenarios so that they always correspond to white moves. The original dataset is extracted from (quantum24, 2023), and an illustrative example is shown in Figure 14.

### B.2. Route Planning

We evaluate route planning in two-dimensional grid environments. The objective across tasks is to construct valid paths that connect designated start and goal locations under different structural constraints. We consider two tasks: *Maze* and *Shortest Path*.

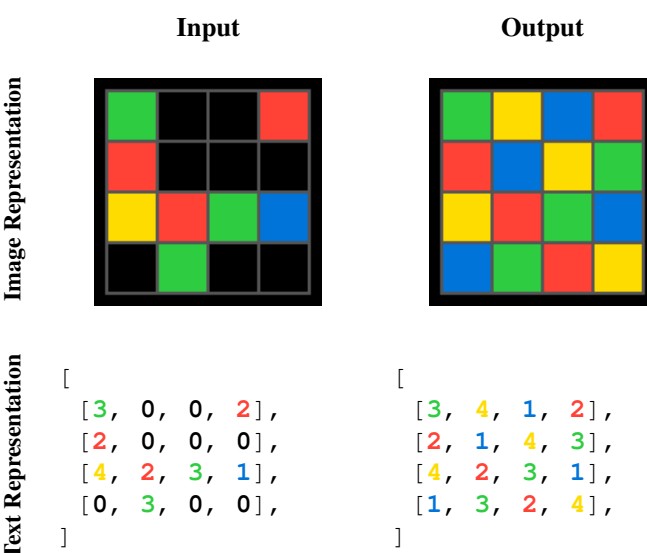

*Figure 11.* Example input-output pair for task *Sudoku Mini*.

### B.2.1. MAZE

**Objective:** Navigate from the start cell to the goal cell through a grid containing blocked and open positions.

**Rules:**

1. The agent starts at the top-left cell and must reach the bottom-right cell.

2. Movement is allowed only through open cells.

3. Allowed moves are up, down, left, and right (no diagonal moves).

4. A valid solution is a continuous sequence of moves from start to goal.

We evaluate two scenarios:

- **Base Maze:** Training and evaluation on $21 \times 21$ grids.

- **Maze Generalization:** Training on smaller $13 \times 13$ grids and testing on larger $21 \times 21$ grids.

We illustrate a sample $21 \times 21$ maze in Figure 16, which serves as training and evaluation data in the *Base Maze* setting and as evaluation data in the *Maze Generalization* setting. Figure 15 shows a sample $13 \times 13$ maze, which is used as training data in the *Maze Generalization* setting.

### B.2.2. SHORTEST PATH

**Objective:** Connect two arbitrary points with the shortest possible route.

**Rules:**

1. Start and goal cells are specified anywhere on the grid.

2. Movement is allowed only through open cells.

3. Allowed moves are up, down, left, and right (no diagonal moves).

**Input**          **Output**

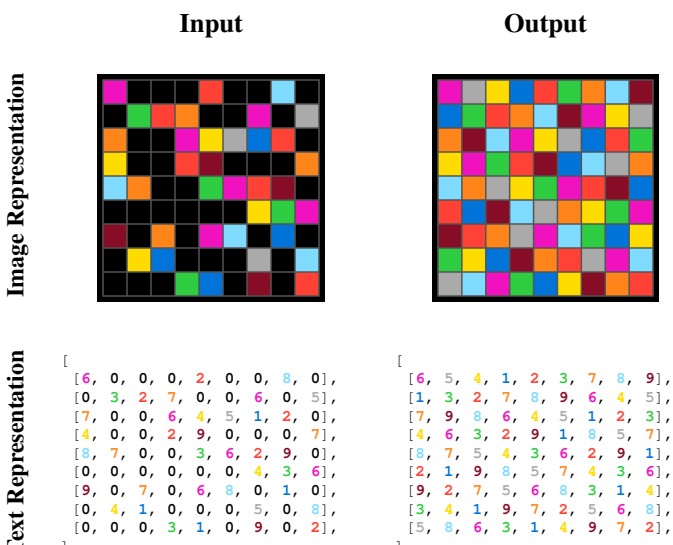

*Figure 12.* Example input-output pair for task *Sudoku*.

4. A valid solution is a continuous path from start to goal with minimal length among all possible paths.

We provide an example in Figure 17.

### B.3. Cellular Automata

#### B.3.1. ELEMENTARY CELLULAR AUTOMATA (ECA)

Elementary Cellular Automata (ECA) are one-dimensional binary-state automata defined on a line of cells. Each cell $c_i^t \in \{0, 1\}$ at time $t$ updates based on itself and its two neighbors:

$$c_i^{t+1} = f(c_{i-1}^t, c_i^t, c_{i+1}^t),$$

where $f$ is specified by a rule number between 0 and 255.

For example, Rule 110 is encoded by the binary string `01101110`, which maps the eight possible neighborhoods $(c_{i-1}^t, c_i^t, c_{i+1}^t)$ to the next state:

| Neighborhood | 111 | 110 | 101 | 100 | 011 | 010 | 001 | 000 |
|---|---|---|---|---|---|---|---|---|
| Next state | 0 | 1 | 1 | 0 | 1 | 1 | 1 | 0 |

We evaluate four representative rules from each of Wolfram's classes (Wolfram, 1984), summarized in Table 8.

Rule 110 is well known for its complex localized structures and universality (Cook, 2004). We show an example in Figure 18.

#### B.3.2. LIFE-LIKE CELLULAR AUTOMATA

Life-like CA generalize Conway's Game of Life (Gardner, 1970), using binary cells on a two-dimensional grid. Each cell updates according to the number of live neighbors in the Moore neighborhood (eight adjacent cells). In standard Game of Life ($B3/S23$):

$$c_{i,j}^{t+1} = \begin{cases} 1 & \text{if cell is dead and has exactly 3 live neighbors (birth),} \\ 1 & \text{if cell is alive and has 2 or 3 live neighbors (survival),} \\ 0 & \text{otherwise (death).} \end{cases}$$

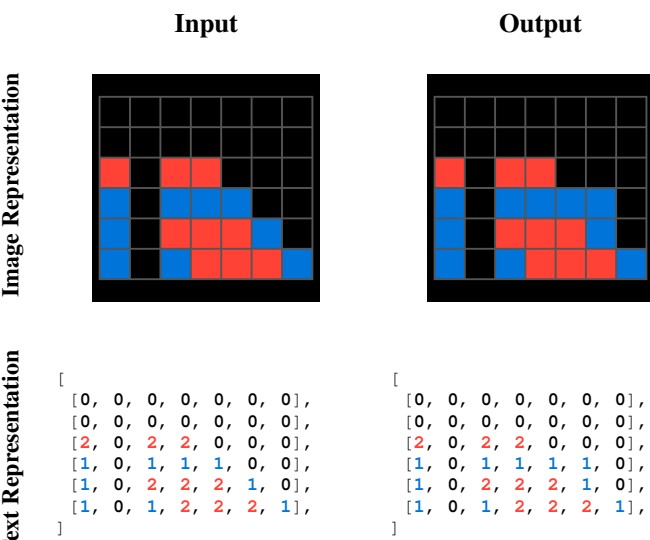

*Figure 13.* Example input-output pair for task *Connect4*.

We consider several well-known Life-like variants. These rules, summarized in Table 9, capture diverse behaviors ranging from explosive growth to symmetry under inversion. We shown an example in Figure 19 of the basic Game of Life.

### B.3.3. LANGTON'S ANT

Langton's ant (Langton, 1986) is an agent-based CA where a single agent moves on a binary grid. At each step:

$$(x, y), d, g(x, y) \rightarrow (x', y'), d', g'(x, y),$$

where $(x, y)$ is the current cell, $d$ is direction, and $g(x, y) \in \{0, 1\}$ is the cell state.

1. If $g(x, y) = 0$, turn right; if $g(x, y) = 1$, turn left.

2. Flip the cell color: $g'(x, y) = 1 - g(x, y)$.

3. Move forward one step.

After many steps, chaotic behavior gives way to a repeating "highway" structure. To make the task predictable, **we always start with the ant facing on the same initial direction and being on top of a 0 cell.** For an example see Figure 20

## C. Additional Qualitative Results

### C.1. ARC-AGI

To further illustrate the complementary strengths of VDMs and LLMs, we include qualitative examples of ARC-AGI tasks. In some cases, the LLM enables it to find the correct solution, while the VDM fails. Examples of this behavior is shown in Figure 23.

In contrast, there are tasks where both models succeed, suggesting that the underlying structure can be captured through either symbolic reasoning or visual pattern learning. One such case is given in Figure 24.

Finally, we highlight situations where only the VDM solves the task correctly (Figures 21 and 22). These examples emphasize how visual inductive biases allow the VDM to generalize in settings where symbolic reasoning alone appears insufficient.

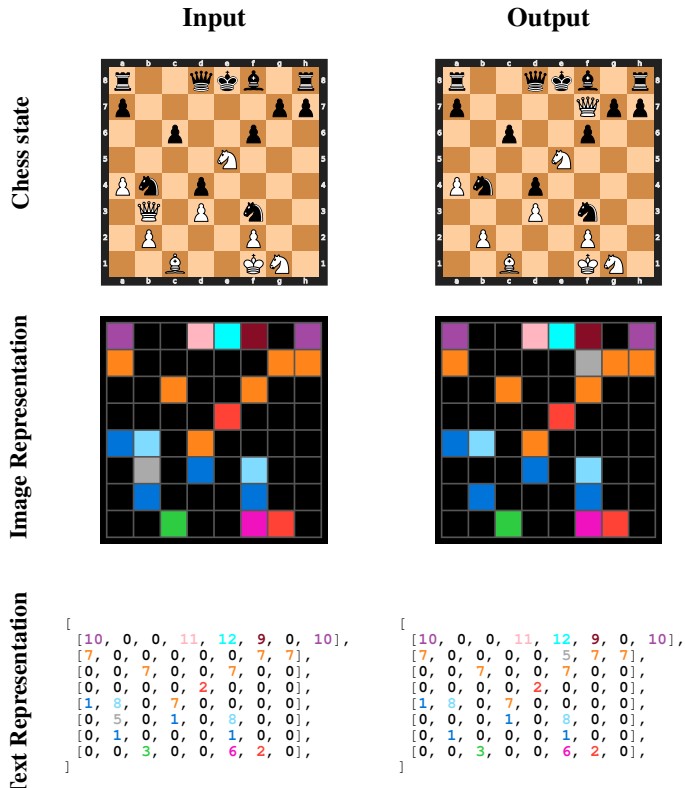

*Figure 14.* Example input-output pair for task *Chess Mate in 1*.

## C.2. Structured Visual Tasks

We include additional qualitative examples from structured visual tasks such as mazes, route planning, and cellular automata, complementing the quantitative results in the main text.

# D. Additional Results

# E. ARC Family

Here, we include the comparison table for ConceptARC, by including finetuned LLMs (Qwen3-4B-Instruct, Qwen3-8B, LLama3.1-8B-Instruct) and GPT-4 [IC][2] (Moskvichev et al., 2023), as well as VDMs (CogVideoX1.5-5B, Wan2.1-14B, LTX-2B/13B). These additional results provide broader context and help reinforce the trends observed in the main text. See Table 10.

The relatively lower performance of LTX compared to other VDMs may stem from its aggressive VAE compression, which can discard structural information important for ConceptARC. This reflects a design tradeoff of the LTX models, aimed at enabling much faster video generation (HaCohen et al., 2025).

## E.1. Pitfalls of Vision Language Models

Vision–Language Models (VLMs) promise to bridge the gap between visual perception and language by training on vast datasets of paired images and text. In principle, this multimodal pretraining should enable these models to solve visually grounded tasks more effectively than language-only models. To test whether this promise holds in practice, we evaluate a

---

[2]Added for reference with commercial models, this case is directly IC and not our finetune approach.

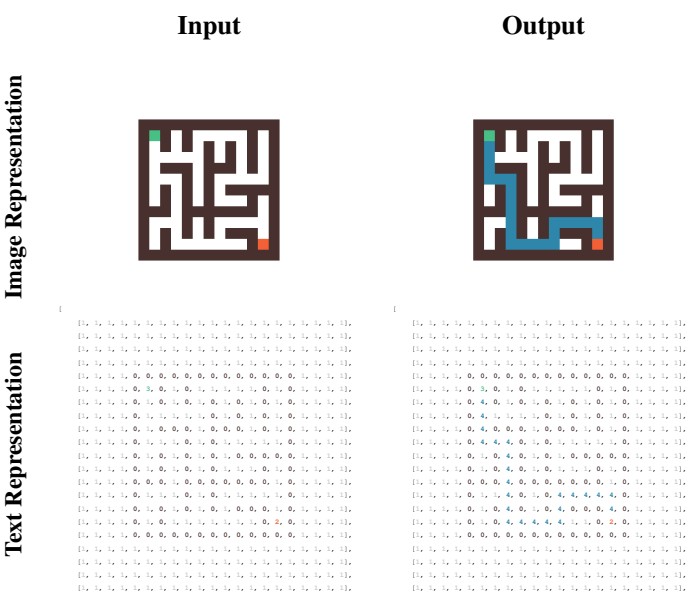

*Figure 15.* Example input-output pair for task *Maze Small*.

representative VLM, Gemma-4B (Gemma Team, 2025), on a structured visual task: *Sudoku.*

We fine-tune the same model with $n = 1000$ samples under three configurations: **text-only**, **image-only**, and **combined image–text**; keeping all other settings fixed. The results in Table 11 reveal a striking limitation: adding image input offers no measurable improvement, and the **image-only** variant performs worse than a trivial baseline. This suggests that the model is unable to extract meaningful information from visual inputs, even when explicitly trained to do so.

To investigate why, we train the **image-only** model on a simplified task: reconstructing the textual grid representation of its own image input rather than predicting a Sudoku solution. With small training sets ($n = 3, 5, 10$), the model fails to interpret the images and instead memorizes training samples, reproducing them verbatim regardless of input (Table 12). The model learns little about the underlying structure of the visual input.

This experiment exposes a deeper issue: despite their multimodal pretraining, current VLMs struggle to extract structured information from images (Jing et al., 2025; Sim et al., 2025). They appear to rely primarily on semantics and basic pattern recognition rather than true visual understanding. Furthermore, VLMs inherit many of the limitations of LLMs, such as reliance on text-based outputs, without gaining meaningful visual understanding ability.

Because VLMs provide no measurable advantage over language-only models for these structured visual tasks, we focus on LLMs as the primary baseline. LLMs already demonstrate strong capabilities in structured prediction and symbolic manipulation, making them a fair and informative comparison point for VDMs. This framing keeps the evaluation focused on model families that offer complementary strengths.

## F. Results - Full Tables

We provide the complete set of experimental results, which constitute the underlying data for the figures reported in the main paper.

## G. Exploring Generalization of I2I-Tuned VDMs

While the main text emphasizes grid-structured visual prediction tasks, our framework extends naturally to a broad range of image-to-image problems. In this section, we briefly explore its applicability to classical computer vision tasks. Few-shot adaptation functions both as an efficient tuning strategy and as a probe of model competence: if the model succeeds with **very few paired examples**, it indicates that the underlying ability was already internalized during pretraining.

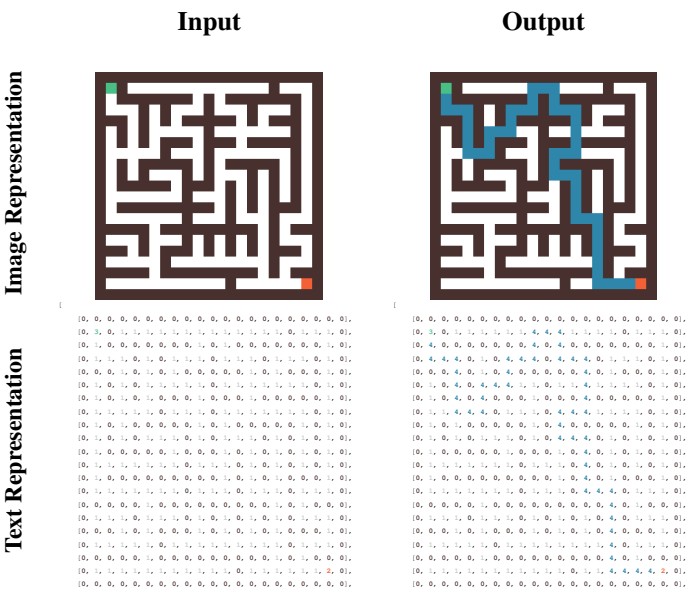

*Figure 16.* Example input-output pair for task *Maze*.

We fine-tune CogVideoX1.5-5B, across tasks using between one and thirty paired examples, maintaining the same architecture, optimization schedule, and hyperparameters as in the main experiments. No auxiliary losses or task-specific modifications are introduced, isolating the contribution of pretrained knowledge.

We explore this setup on several established datasets spanning diverse visual domains, including **NYUv2** (Silberman et al., 2012), **ADE20K** (Zhou et al., 2017; 2019), **ML-Hypersim** (Roberts et al., 2021), **COCO 2017** (Lin et al., 2014), and **DreamBooth** (Ruiz et al., 2022). These benchmarks cover a wide range of classical computer vision problems, from structured scene understanding to generative image transformation.

Figure 30 illustrates that the model can capture geometric transformations under extreme few-shot conditions. We further show one-shot style transfer in Figure 31.

We also qualitative show this framework can be used to solve some classical computer vision tasks. In Figure 33 we show examples after training with only $n = 30$ samples for *Binary Segmentation* for dogs and *Pose* estimation for humans.

**Input**                              **Output**

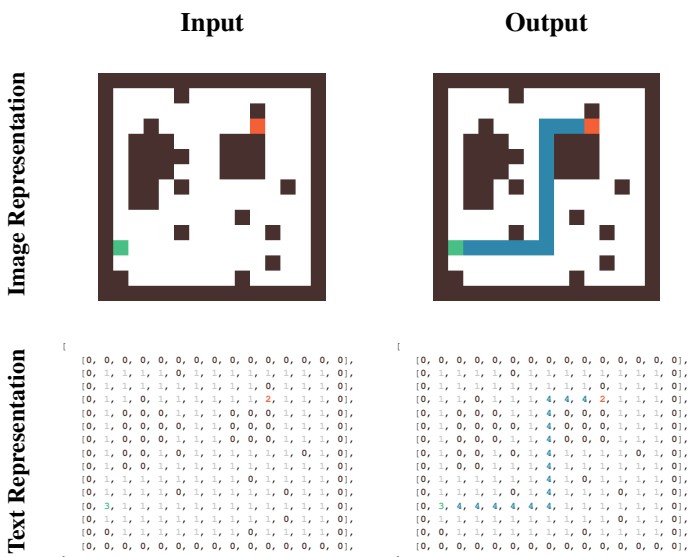

*Figure 17.* Example input-output pair for task *Shortest Path*.

*Table 8.* Representative Elementary Cellular Automata rules by Wolfram class.

| Class | Rules |
|-------|-------|
| Class 1 | 8, 32, 128, 160 |
| Class 2 | 4, 108, 170, 250 |
| Class 3 | 30, 45, 90, 150 |
| Class 4 | 110, 54, 62, 106 |

**Input**                              **Output**

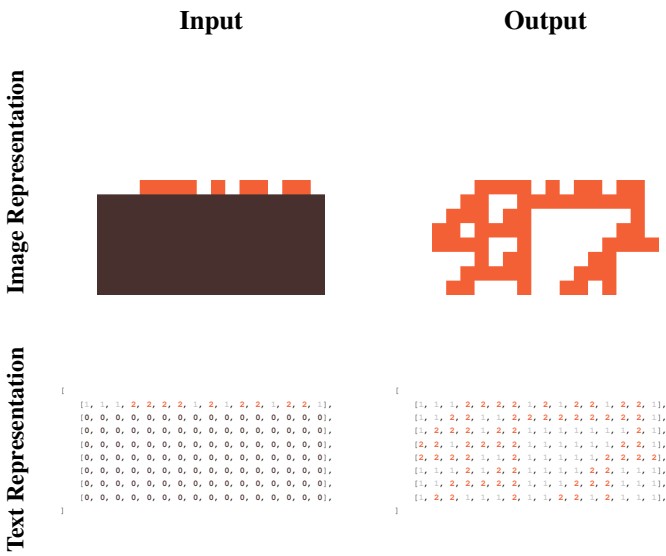

*Figure 18.* Example input-output pair for task *Langton ant step 2*.

**Input**          **Output**

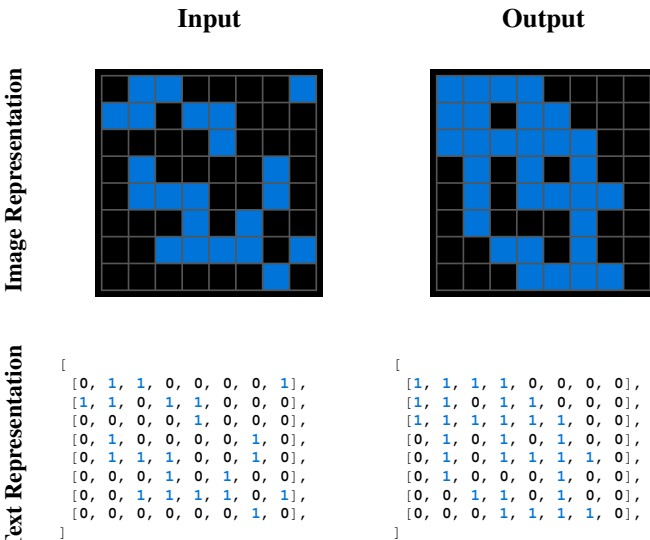

*Figure 19.* Example input-output pair for task *Game of Life step 1*.

*Table 9.* Life-like cellular automata variants evaluated.

| Name | Rule (B/S) | Description |
|------|-----------|-------------|
| Day & Night | B3678/S34678 | Symmetric under inversion; complex dynamics |
| Maze | B3/S12345 | Generates labyrinth-like, maze-like growth |
| Seeds | B2/S∅ | All live cells die each step; explosive expansion |
| Life | B3/S2 | Sparse survival; promotes small, mobile clusters |

**Input**          **Output**

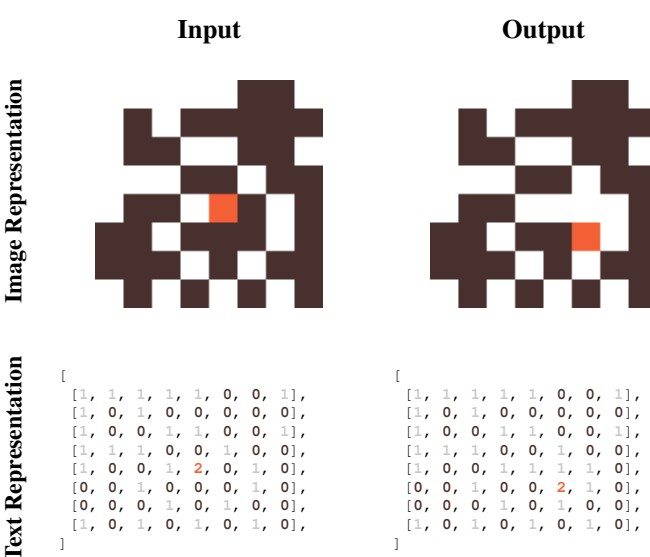

*Figure 20.* Example input-output pair for task *Langton ant step 2*.

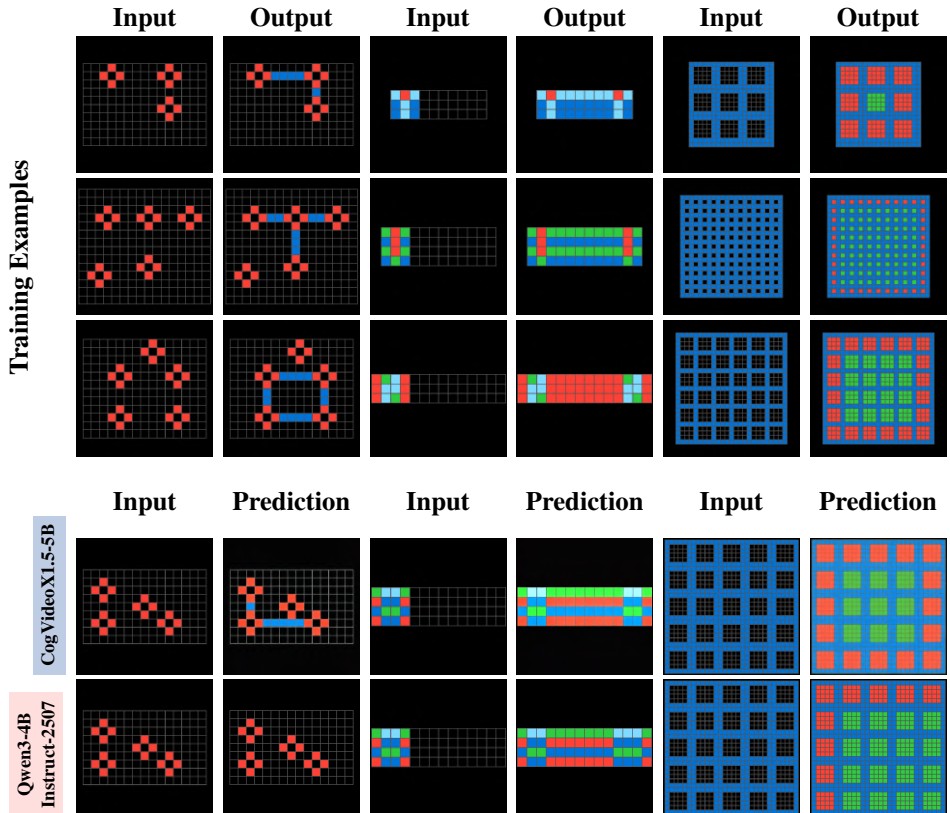

*Figure 21.* Qualitative results on ARC-AGI for problems *60a26a3e*, *62b74c02*, *8a371977*.

*Table 10.* Concept-wise overall accuracy across models. Best values are highlighted for **VDMs** or **LLMs** .

| Concept | LTX-13B | LTX-2B | Wan2.1-14B | CogVideoX1.5-5B | Qwen3-4B Instruct-2507 | Qwen3-8B | Llama3.1-8B | GPT-4 [IC] |
|---|---|---|---|---|---|---|---|---|
| AboveBelow | 0.30 | 0.17 | 0.37 | **0.40** | **0.40** | **0.40** | 0.17 | 0.23 |
| TopBottom2D | 0.23 | 0.17 | **0.63** | 0.37 | 0.50 | 0.50 | 0.37 | 0.23 |
| TopBottom3D | 0.27 | 0.17 | **0.47** | 0.33 | 0.13 | 0.20 | 0.17 | 0.20 |
| HorizontalVertical | 0.13 | 0.20 | **0.53** | 0.47 | 0.43 | 0.47 | 0.33 | 0.27 |
| Center | 0.33 | 0.30 | **0.57** | 0.37 | 0.20 | 0.20 | 0.13 | 0.33 |
| FilledNotFilled | 0.30 | 0.27 | **0.50** | 0.37 | 0.27 | 0.23 | 0.20 | 0.17 |
| CompleteShape | 0.20 | 0.10 | **0.40** | 0.37 | 0.23 | 0.30 | 0.13 | 0.23 |
| InsideOutside | 0.27 | 0.27 | **0.37** | 0.33 | 0.13 | 0.20 | 0.13 | 0.10 |
| ExtractObjects | 0.07 | 0.07 | **0.23** | 0.07 | 0.10 | 0.10 | 0.03 | 0.03 |
| Count | 0.40 | 0.43 | **0.83** | 0.57 | 0.13 | 0.13 | 0.17 | 0.13 |
| SameDifferent | 0.23 | 0.23 | 0.33 | **0.37** | 0.27 | 0.23 | 0.27 | 0.17 |
| Order | 0.03 | 0.03 | 0.00 | 0.07 | **0.27** | **0.27** | 0.10 | **0.27** |
| MoveToBoundary | 0.17 | 0.00 | 0.13 | 0.17 | **0.23** | 0.10 | 0.17 | 0.20 |
| ExtendToBoundary | 0.20 | 0.23 | **0.50** | 0.40 | 0.13 | 0.17 | 0.10 | 0.07 |
| Copy | 0.20 | 0.03 | 0.17 | 0.13 | 0.17 | 0.10 | 0.10 | **0.23** |
| CleanUp | 0.43 | 0.40 | **0.60** | 0.53 | 0.27 | 0.30 | 0.27 | 0.20 |
| **Average Accuracy** | 0.24 | 0.19 | **0.41** | 0.33 | 0.24 | 0.24 | 0.18 | 0.19 |

*Table 11.* Relative Accuracy and Accuracy on *Sudoku*.

| Model | Relative Accuracy | Accuracy |
|---|---|---|
| **Text-only** | 0.79 | 0.06 |
| **Combined image–text** | 0.78 | 0.06 |
| **Image-only** | 0.12 | 0.00 |

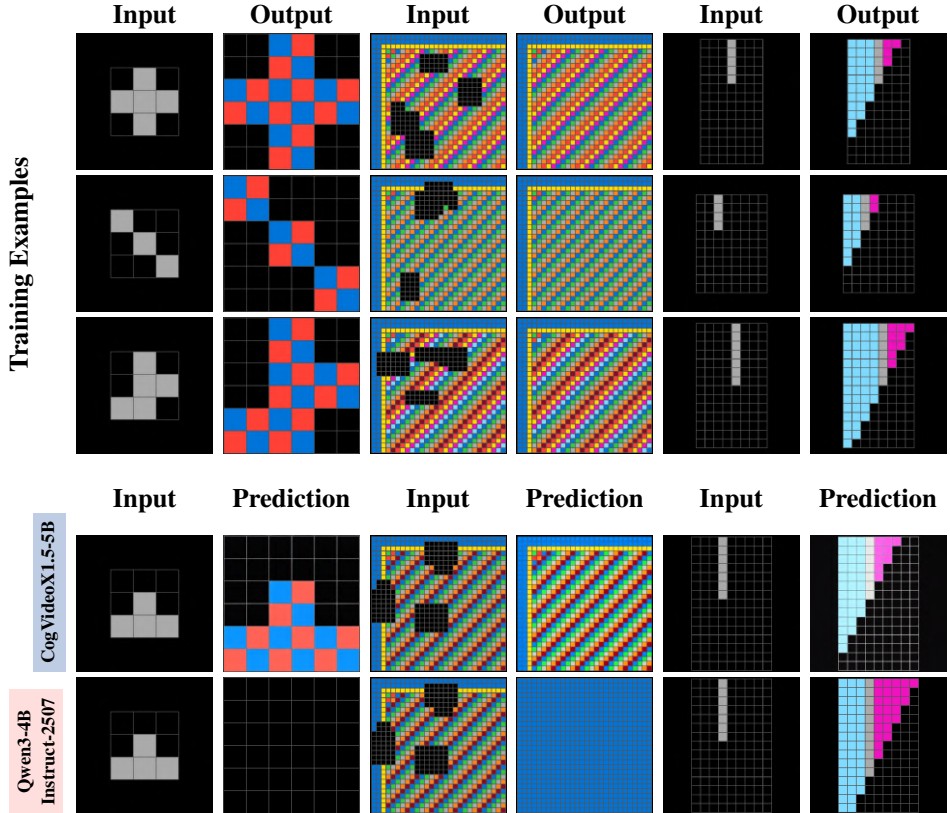

*Figure 22.* Qualitative results on ARC-AGI for problems *2072aba6*, *4aab4007*, *5207a7b5*.

*Table 12.* Distribution of outputs on the test set exactly matching training samples for different training set sizes.

| Training Set Size | Sample | Proportion | Total Proportion |
|---|---|---|---|
| 3 | Sample 1 | 0.385 | **1.00** |
| | Sample 2 | 0.010 | |
| | Sample 3 | 0.605 | |
| 5 | Sample 1 | 0.490 | **0.99** |
| | Sample 2 | 0.030 | |
| | Sample 3 | 0.335 | |
| | Sample 4 | 0.135 | |
| 10 | Sample 1 | 0.100 | **0.96** |
| | Sample 2 | 0.010 | |
| | Sample 3 | 0.030 | |
| | Sample 4 | 0.005 | |
| | Sample 5 | 0.015 | |
| | Sample 6 | 0.170 | |
| | Sample 7 | 0.615 | |
| | Sample 8 | 0.010 | |

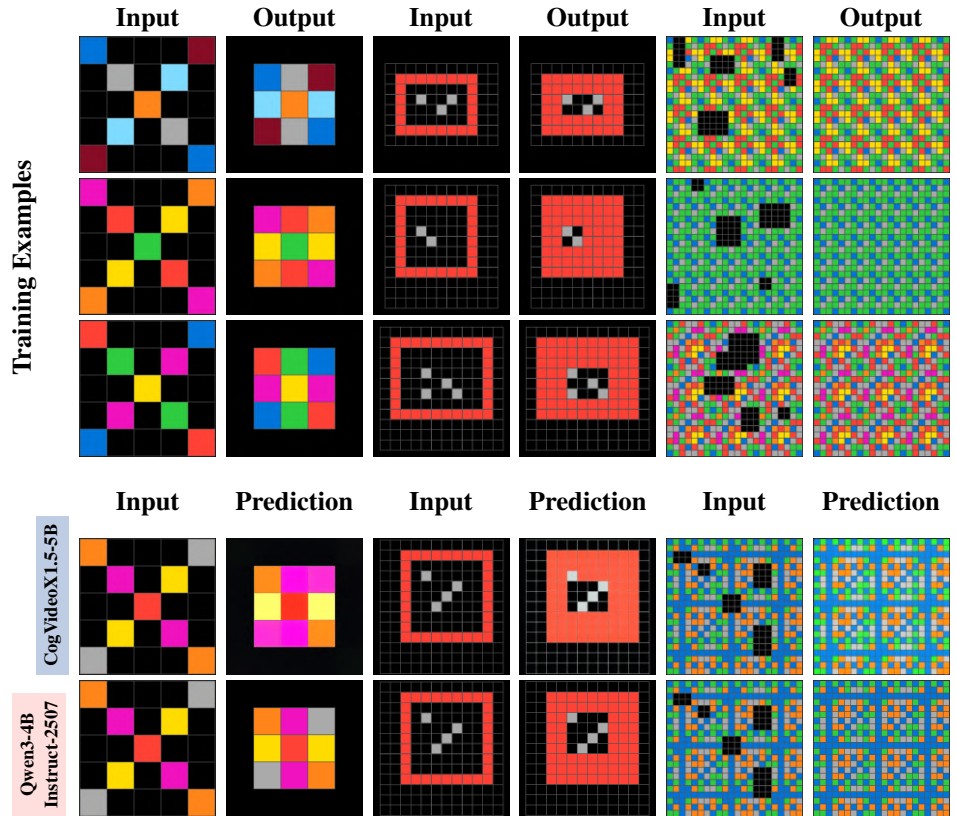

*Figure 23.* Qualitative results on ARC-AGI for problems *ca8de6ea*, *d37a1ef5*, *e95e3d8e*.

*Table 13.* Comparison of CogVideoX1.5-5B and Qwen3-4B-Instruct-2507 accuracy on structured games. Missing values are shown as −.

| n | CogVideoX1.5-5B | | | | | Qwen3-4B-Instruct-2507 | | | | |
|---|---|---|---|---|---|---|---|---|---|---|
| | Chess-Mate-in-1 | Connect 4 | Hitori 5x5 | Sudoku Mini | Sudoku | Chess-Mate-in-1 | Connect 4 | Hitori 5x5 | Sudoku Mini | Sudoku |
| 3 | 0.00 | 0.44 | 0.01 | 0.22 | 0.00 | 0.00 | 0.03 | 0.00 | 0.18 | − |
| 5 | 0.00 | 0.62 | 0.02 | 0.36 | 0.00 | 0.02 | 0.05 | 0.00 | 0.22 | − |
| 10 | 0.00 | 0.74 | 0.62 | 0.65 | 0.00 | 0.04 | 0.08 | 0.02 | 0.48 | − |
| 30 | 0.02 | 0.78 | 0.72 | 0.78 | 0.20 | 0.13 | 0.38 | 0.02 | 0.64 | 0.00 |
| 50 | 0.04 | 0.80 | 0.84 | 0.90 | 0.34 | 0.15 | 0.38 | 0.10 | 0.68 | 0.00 |
| 100 | 0.08 | 0.85 | 0.92 | 0.91 | 0.60 | 0.24 | 0.69 | 0.28 | 0.78 | 0.01 |
| 300 | 0.14 | 0.84 | 0.94 | 0.90 | 0.55 | 0.38 | 0.71 | 0.57 | 0.80 | 0.01 |
| 500 | 0.20 | 0.89 | 0.94 | 0.94 | 0.60 | 0.44 | 0.69 | 0.64 | 0.86 | 0.06 |
| 1000 | 0.22 | 0.90 | 0.96 | 0.91 | 0.79 | 0.56 | 0.76 | 0.86 | 0.90 | 0.14 |
| 3000 | − | 0.92 | 0.98 | 0.95 | 0.86 | − | 0.78 | 0.94 | 0.92 | 0.32 |
| 5000 | − | 0.90 | 0.99 | 0.96 | 0.86 | − | 0.82 | 0.96 | 0.96 | 0.55 |

*Table 14.* Comparison of CogVideoX1.5-5B and Qwen3-4B-Instruct-2507 accuracy on Life-Like Cellular Automata variants. Missing values are shown as −.

| n | CogVideoX1.5-5B | | | | | Qwen3-4B-Instruct-2507 | | | | |
|---|---|---|---|---|---|---|---|---|---|---|
| | Life_B3S2 | DayAndNight | Maze | Seeds | Game of Life | Life_B3S2 | DayAndNight | Maze | Seeds | Game Of Life |
| 10 | 0.00 | 0.00 | 0.00 | 0.00 | 0.00 | − | − | − | − | − |
| 30 | 1.00 | 0.81 | 0.87 | 1.00 | 0.96 | − | 0.63 | 0.81 | 0.75 | 0.63 |
| 50 | 1.00 | 0.95 | 0.91 | 1.00 | 0.97 | − | 0.64 | 0.80 | 0.78 | 0.64 |
| 100 | 1.00 | 1.00 | 0.96 | 1.00 | 1.00 | 0.61 | 0.70 | 0.87 | 0.63 | 0.73 |
| 300 | − | − | − | − | − | 1.00 | 1.00 | 1.00 | 1.00 | 1.00 |
| 500 | − | − | − | − | − | − | 1.00 | 1.00 | 1.00 | 1.00 |

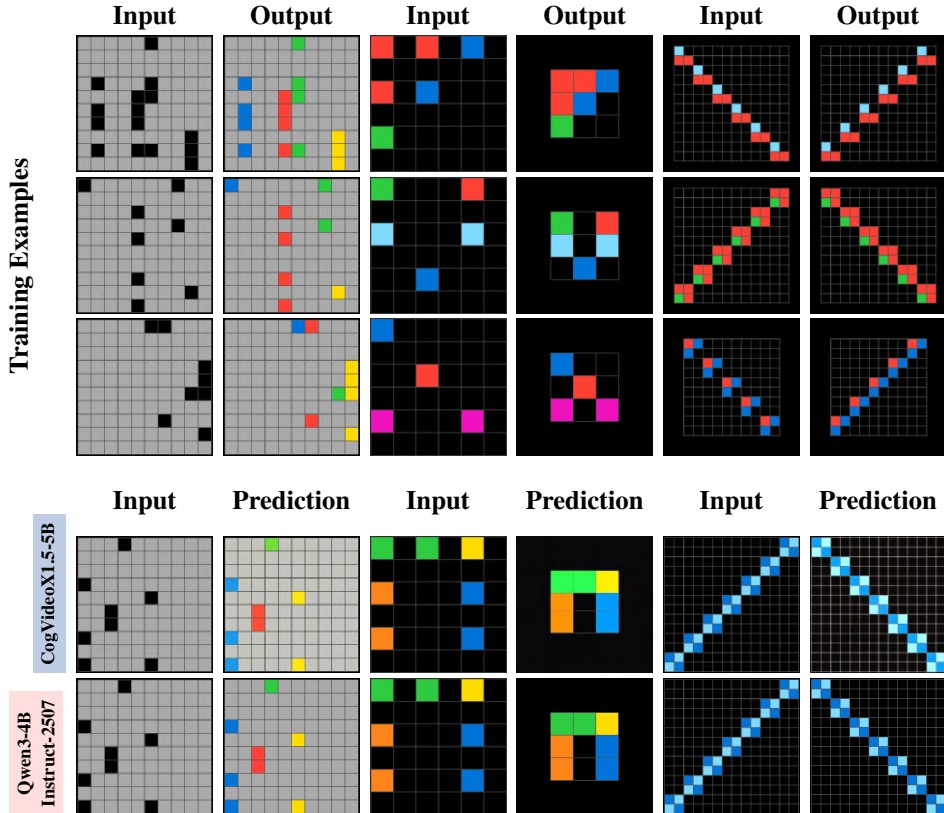

*Figure 24.* Qualitative results on ARC-AGI for problems *575b1a71*, *68b67ca3*, *8ee62060*.

*Table 15.* Comparison of CogVideoX1.5-5B and Qwen3-4B-Instruct-2507 accuracy on Langton's Ant with respect to number of steps into the future. Missing values are shown as −.

| n | CogVideoX1.5-5B | | | | Qwen3-4B-Instruct-2507 | | | |
|---|---|---|---|---|---|---|---|---|
| | Step 2 | Step 3 | Step 5 | Step 10 | Step 2 | Step 3 | Step 5 | Step 10 |
| 3 | 0.18 | 0.03 | 0.03 | – | 0.32 | 0.03 | – | – |
| 5 | 0.23 | 0.07 | 0.04 | 0.00 | 0.21 | 0.04 | – | – |
| 10 | 0.67 | 0.29 | 0.06 | 0.01 | 0.51 | 0.19 | – | – |
| 30 | 1.00 | 0.76 | 0.25 | 0.01 | 0.79 | 0.46 | 0.06 | 0.00 |
| 50 | 1.00 | 0.99 | 0.41 | 0.01 | 0.950 | 0.58 | 0.14 | 0.010 |
| 100 | 1.00 | 1.000 | 0.88 | 0.08 | 0.99 | 0.910 | 0.39 | 0.01 |
| 300 | – | – | 1.00 | 0.42 | 1.00 | 1.00 | 0.98 | 0.12 |
| 500 | – | – | 1.00 | 0.83 | 1.00 | 1.00 | 1.00 | 0.21 |
| 1000 | – | – | 1.00 | 0.98 | 1.00 | 1.00 | 1.00 | 0.47 |
| 3000 | – | – | – | 0.99 | – | – | – | 0.71 |
| 5000 | – | – | – | – | – | – | – | 0.93 |

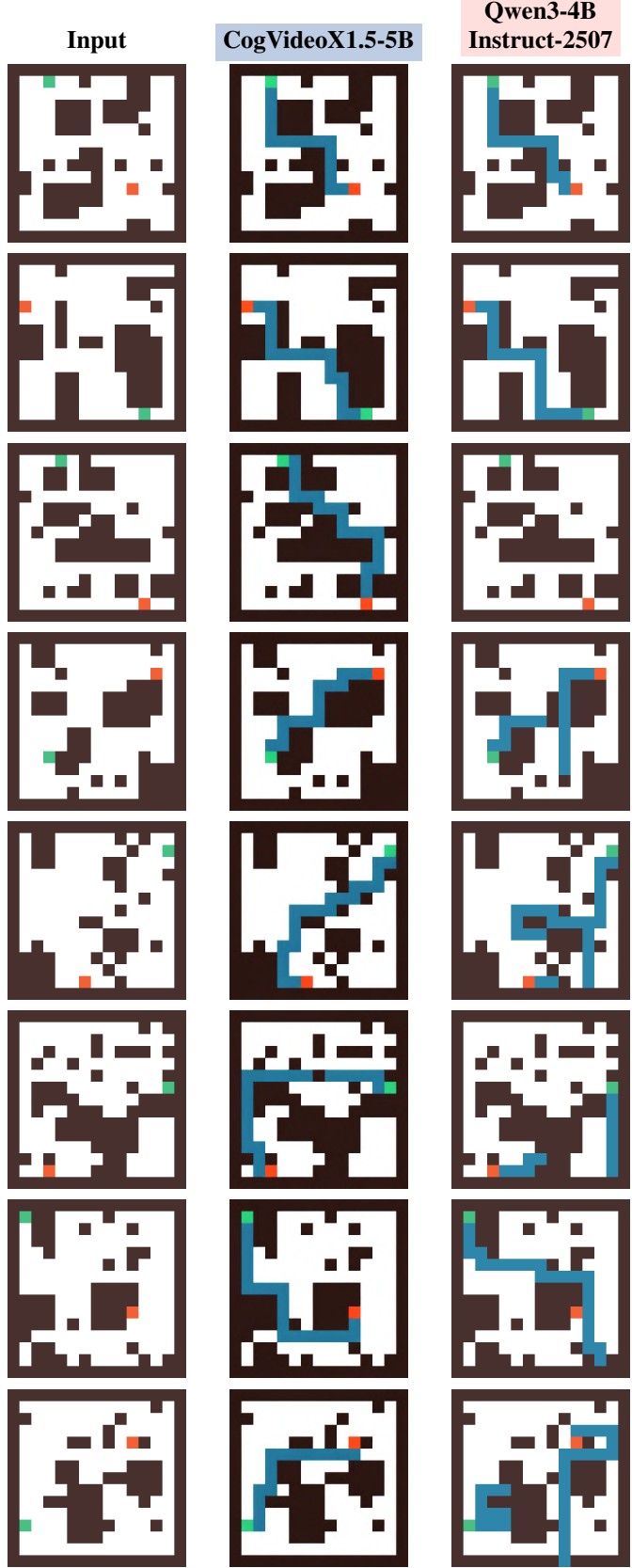

*Figure 25.* Representative examples for the *Shortest Path* task, showing ground truth inputs (left) and model predictions (center and right) after finetuning with $n = 300$ samples.

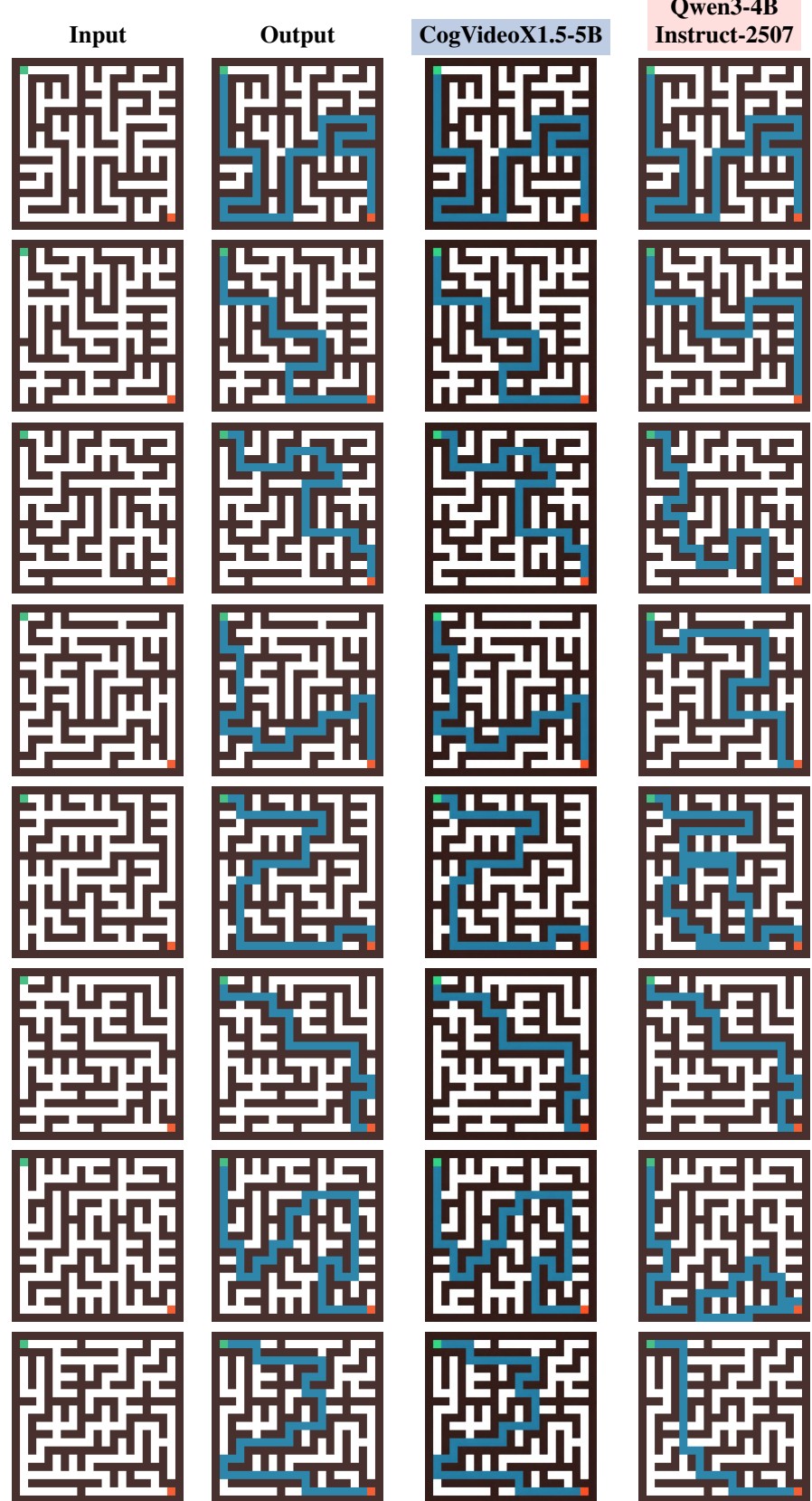

*Figure 26.* Additional qualitative examples for the *Maze* task, showing inputs, ground truth outputs, and model predictions after finetuning with $n = 300$ samples.

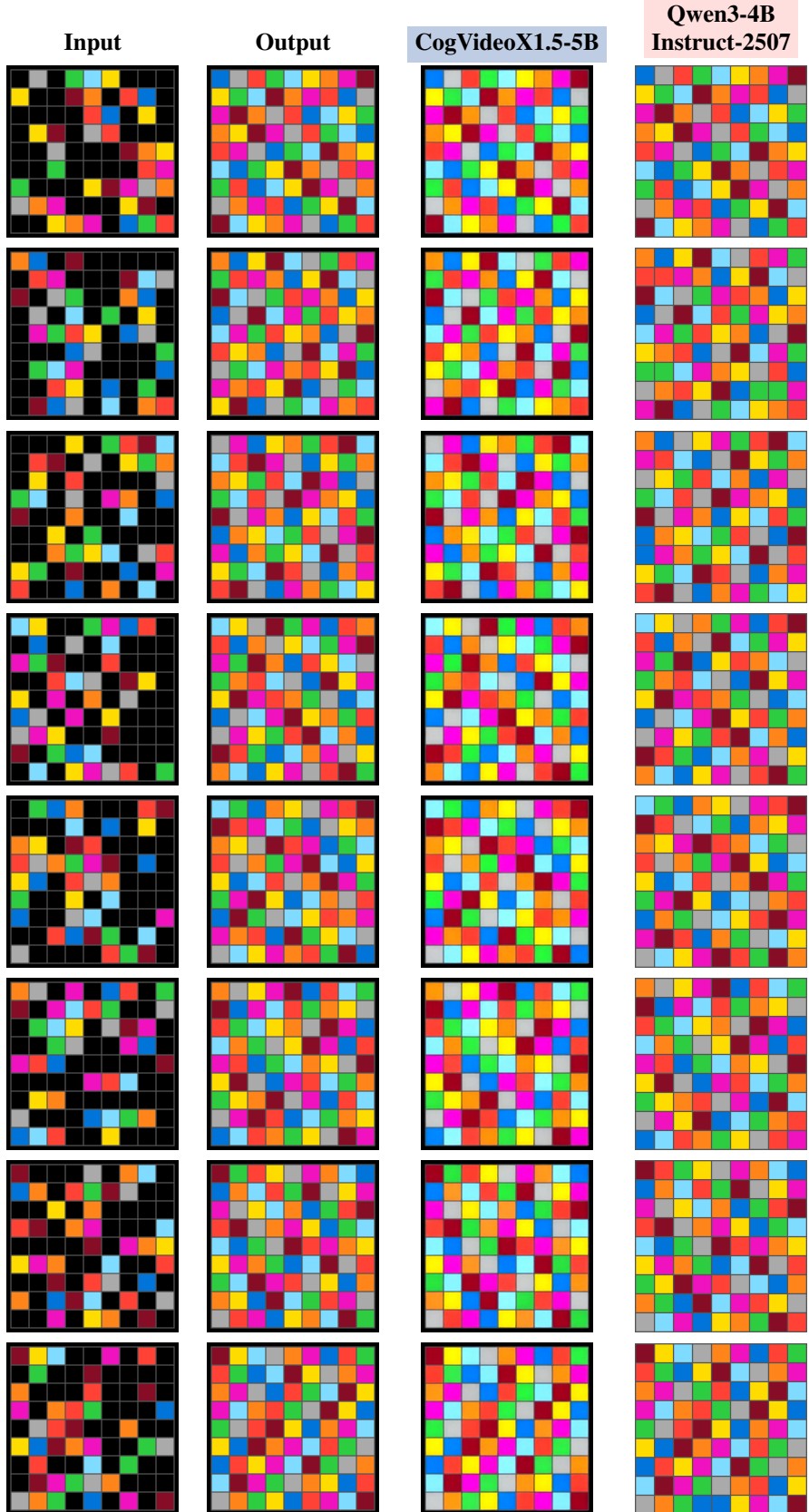

*Figure 27.* Additional qualitative examples for the *Sudoku* task, showing inputs, ground truth outputs, and model predictions after finetuning with $n = 1000$ samples.

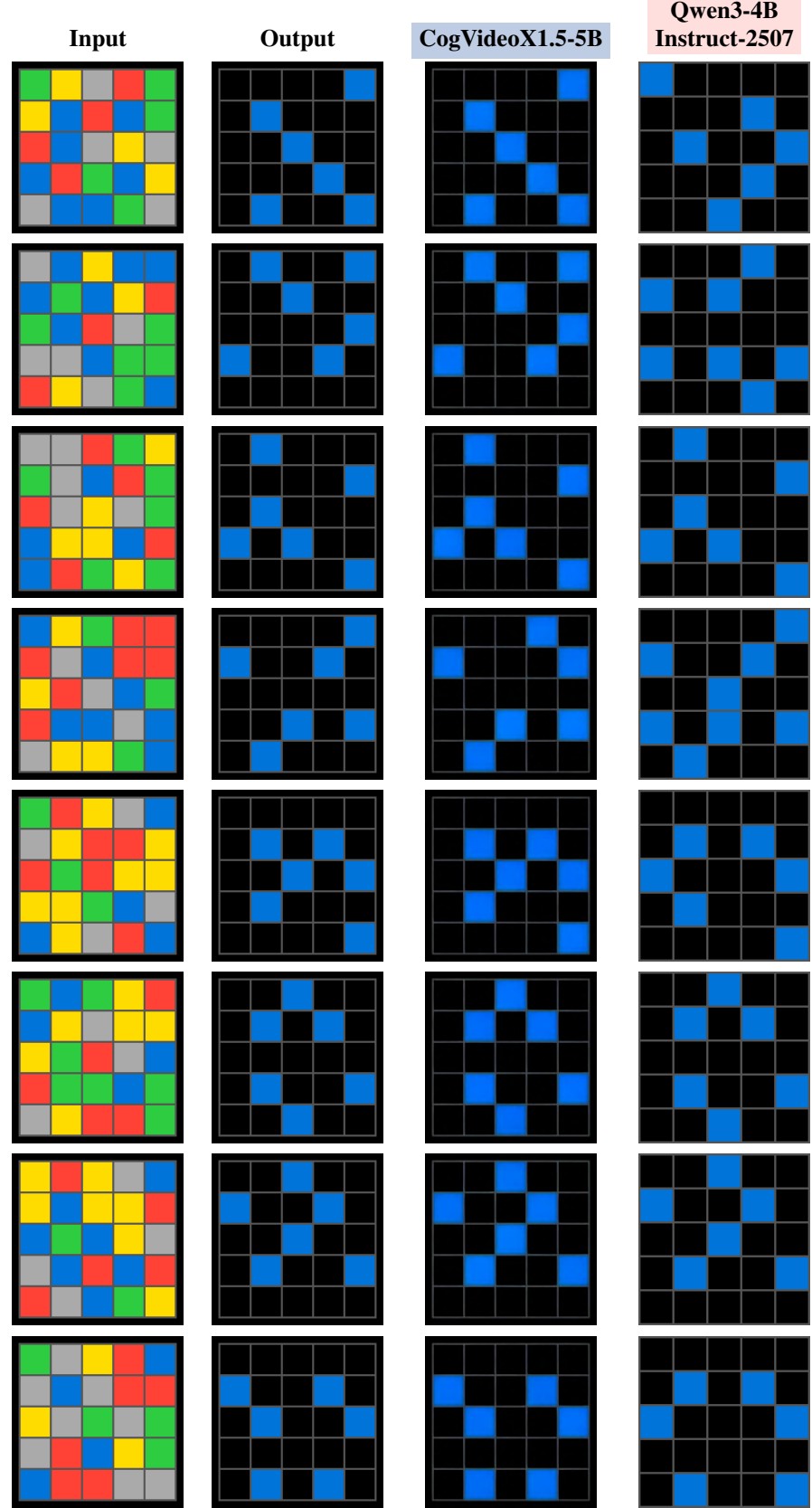

*Figure 28.* Additional qualitative examples for the *Hitori* task, showing inputs, ground truth outputs, and model predictions after finetuning with $n = 100$ samples.

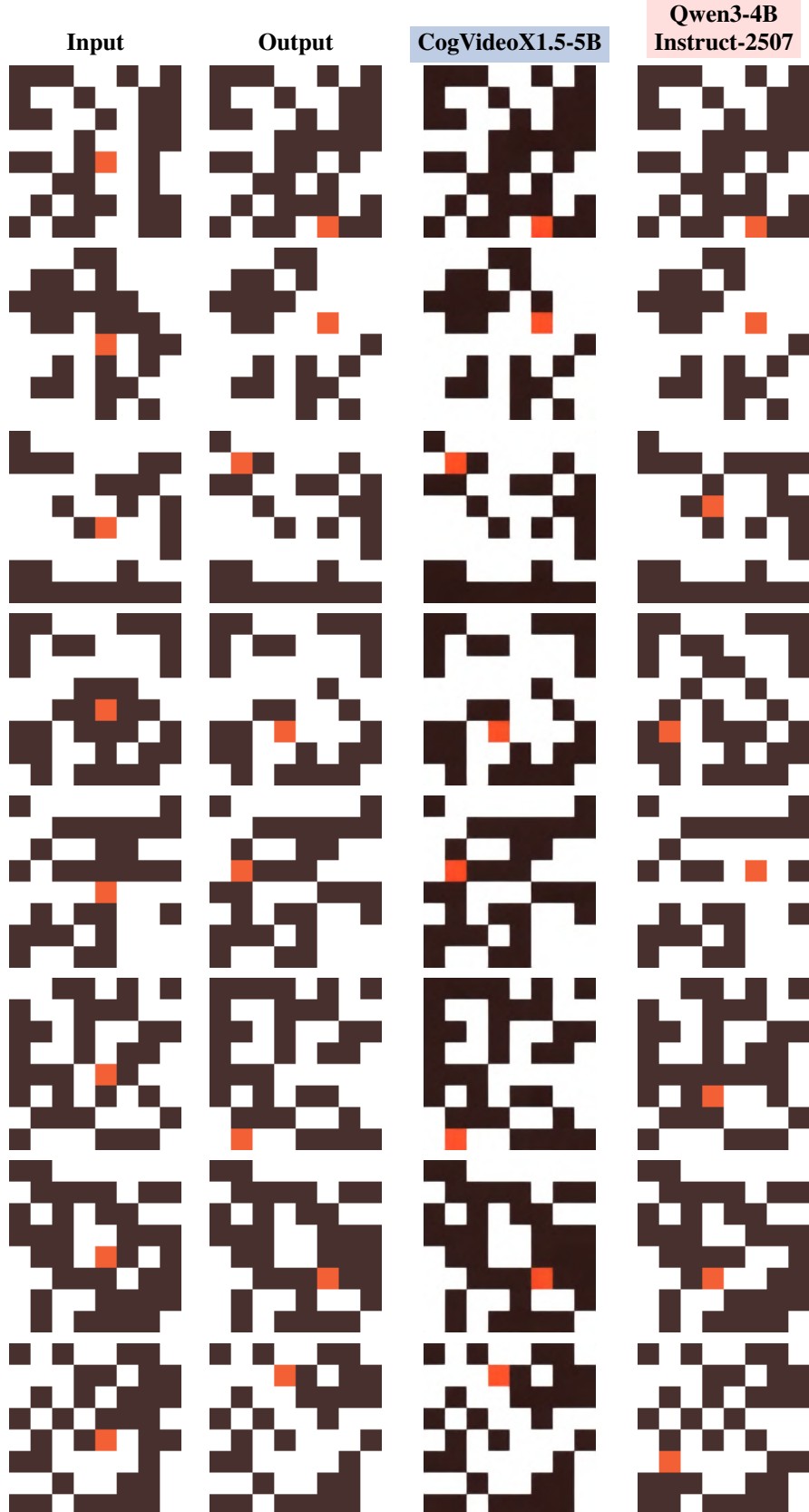

*Figure 29.* Additional qualitative examples for the *Langton Ant* (horizon 10) task, showing inputs, ground truth outputs, and model predictions after finetuning with $n = 1000$ samples.

*Table 16.* Comparison of CogVideoX1.5 and Qwen3-4B-Instruct-2507 accuracy on *Maze* and *Shortest Path* tasks. Missing values are shown as −.

| n | CogVideoX1.5 | | | Qwen3-4B-Instruct-2507 | | |
|---|---|---|---|---|---|---|
| | Base Maze | Maze Generalization | Shortest Path | Base Maze | Maze Generalization | Shortest Path |
| 3 | 0.015 | − | 0.010 | − | − | − |
| 5 | 0.010 | − | 0.025 | − | − | − |
| 10 | 0.070 | 0.050 | 0.040 | − | − | − |
| 30 | 0.550 | 0.175 | 0.330 | 0.000 | − | 0.010 |
| 50 | 0.760 | 0.355 | 0.420 | 0.005 | 0.000 | 0.010 |
| 100 | 0.940 | 0.590 | 0.700 | 0.005 | 0.000 | 0.050 |
| 300 | 1.000 | 0.755 | 0.860 | 0.115 | 0.020 | 0.155 |
| 500 | 1.000 | 0.885 | 0.910 | 0.195 | 0.060 | 0.320 |
| 1000 | − | 0.865 | 0.945 | 0.500 | 0.335 | 0.500 |
| 3000 | − | 0.815 | 0.960 | 0.710 | 0.375 | 0.640 |
| 5000 | − | 0.940 | 0.975 | 0.925 | 0.525 | 0.770 |

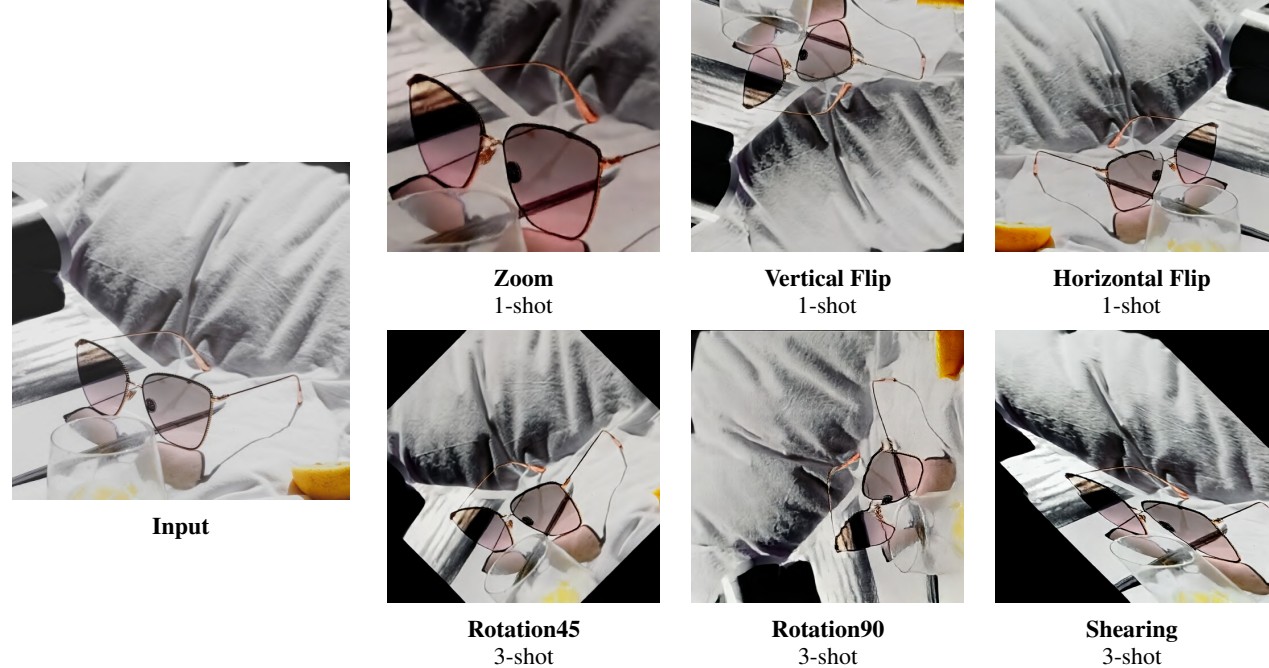

**Input**

**Zoom**
1-shot

**Vertical Flip**
1-shot

**Horizontal Flip**
1-shot

**Rotation45**
3-shot

**Rotation90**
3-shot

**Shearing**
3-shot

*Figure 30.* Geometric transformations learned in few-shot setting. Input is shown on the left, with 1-shot results on the top row and 3-shot results on the bottom row.

*Table 17.* Comparison of CogVideoX1.5-5B and Qwen3-4B-Instruct-2507 accuracy on cellular automata rules grouped by Wolfram classes. Missing values are shown as –.

| n | CogVideoX1.5-5B | | | | Qwen3-4B-Instruct-2507 | | | |
|---|---|---|---|---|---|---|---|---|
| | **Class 1** | | | | | | | |
| | **R8** | **R32** | **R128** | **R160** | **R8** | **R32** | **R128** | **R160** |
| 3 | 0.75 | 0.49 | 0.29 | 0.13 | 0.06 | 0.02 | 0.04 | 0.04 |
| 5 | 0.71 | 0.51 | 0.28 | 0.20 | 0.10 | 0.06 | 0.06 | 0.04 |
| 10 | 0.74 | 0.67 | 0.32 | 0.48 | 0.19 | 0.21 | 0.08 | 0.12 |
| 30 | 0.77 | 0.82 | 0.85 | 0.87 | 0.72 | 0.67 | 0.65 | 0.81 |
| 50 | 0.72 | 0.98 | 0.99 | 0.93 | 0.81 | 0.96 | 0.77 | 0.84 |
| 100 | 1.00 | – | – | – | 0.97 | 0.93 | 0.90 | 0.99 |
| 300 | – | – | – | – | 0.98 | – | – | – |
| | **Class 2** | | | | | | | |
| | **R4** | **R108** | **R170** | **R250** | **R4** | **R108** | **R170** | **R250** |
| 3 | 0.71 | 0.155 | 0.07 | 0.17 | – | – | – | – |
| 5 | 0.76 | 0.310 | 0.27 | 0.19 | – | – | – | – |
| 10 | 0.74 | 0.415 | 0.87 | 0.27 | – | – | 0.85 | – |
| 30 | 0.85 | 0.640 | 1.00 | 0.59 | 0.72 | 0.47 | 0.99 | 0.52 |
| 50 | 0.93 | 0.785 | 1.00 | 0.90 | 0.82 | 0.82 | 0.98 | 0.86 |
| 100 | – | – | – | – | 0.90 | 0.90 | 1.00 | 1.00 |
| 300 | – | – | – | – | 1.00 | 1.00 | 1.00 | 0.99 |
| | **Class 3** | | | | | | | |
| | **R30** | **R45** | **R90** | **R150** | **R30** | **R45** | **R90** | **R150** |
| 3 | 0.00 | 0.00 | 0.00 | 0.00 | – | – | – | – |
| 5 | 0.00 | 0.00 | 0.00 | 0.00 | – | – | – | – |
| 10 | 0.00 | 0.00 | 0.00 | 0.00 | – | – | – | – |
| 30 | 0.07 | 0.07 | 0.10 | 0.00 | 0.18 | 0.03 | 0.03 | 0.01 |
| 50 | 0.55 | 0.53 | 0.25 | 0.01 | 0.83 | 0.71 | 0.08 | 0.97 |
| 100 | 0.97 | 1.00 | 0.99 | 0.65 | 0.97 | 0.98 | 0.27 | 0.99 |
| 300 | – | – | – | 0.86 | 1.00 | 1.00 | 0.90 | 1.00 |
| 500 | – | – | – | 0.98 | – | – | – | – |
| | **Class 4** | | | | | | | |
| | **R110** | **R54** | **R62** | **R106** | **R110** | **R54** | **R62** | **R106** |
| 3 | 0.00 | 0.00 | 0.02 | 0.00 | – | – | – | – |
| 5 | 0.00 | 0.00 | 0.02 | 0.00 | – | – | – | – |
| 10 | 0.00 | 0.01 | 0.03 | 0.00 | – | – | – | – |
| 30 | 0.42 | 0.54 | 0.31 | 0.09 | 0.87 | 0.31 | 0.13 | 0.18 |
| 50 | 0.90 | 0.99 | 0.53 | 0.57 | 0.95 | 0.78 | 0.79 | 0.63 |
| 100 | 1.00 | 1.00 | 0.97 | 0.97 | 1.00 | 0.94 | 0.93 | 1.00 |
| 300 | 1.00 | 1.00 | – | 1.00 | 1.00 | 1.00 | 1.00 | 1.00 |

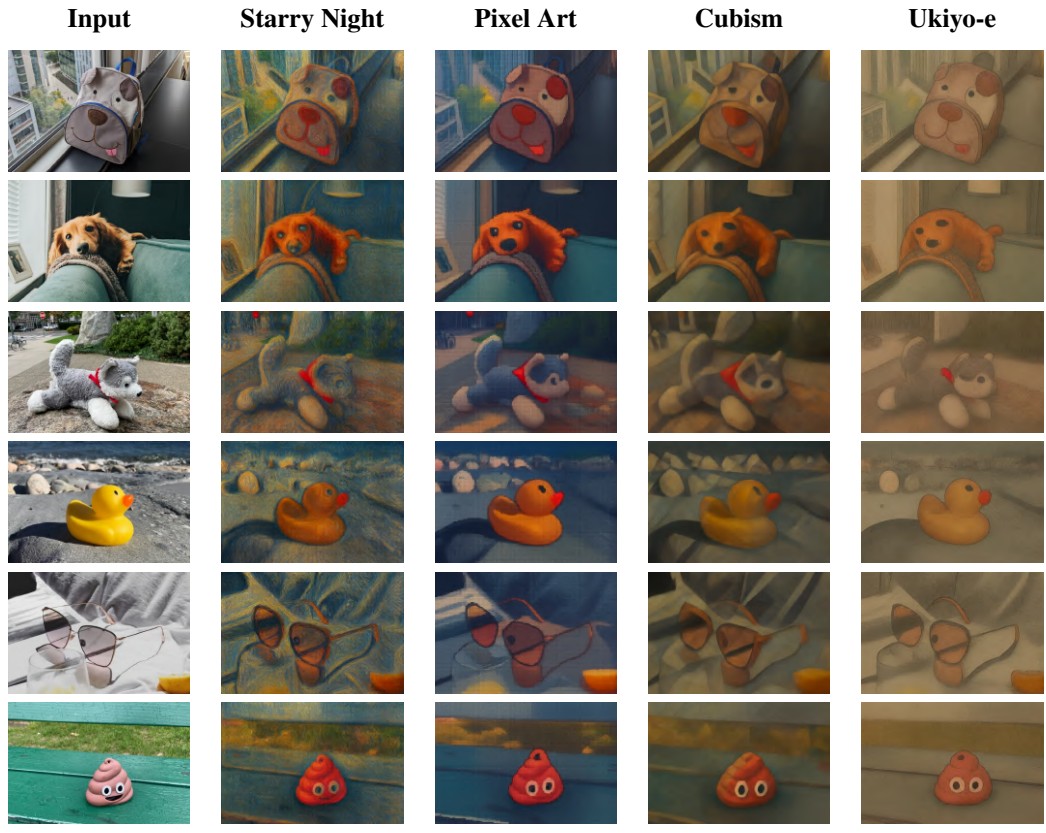

*Figure 31.* 1-shot style transfer results. The model adapts the input images to distinct artistic styles (*Starry Night*, *Pixel Art*, *Cubism*, and *Ukiyo-e*) using only a single reference example.

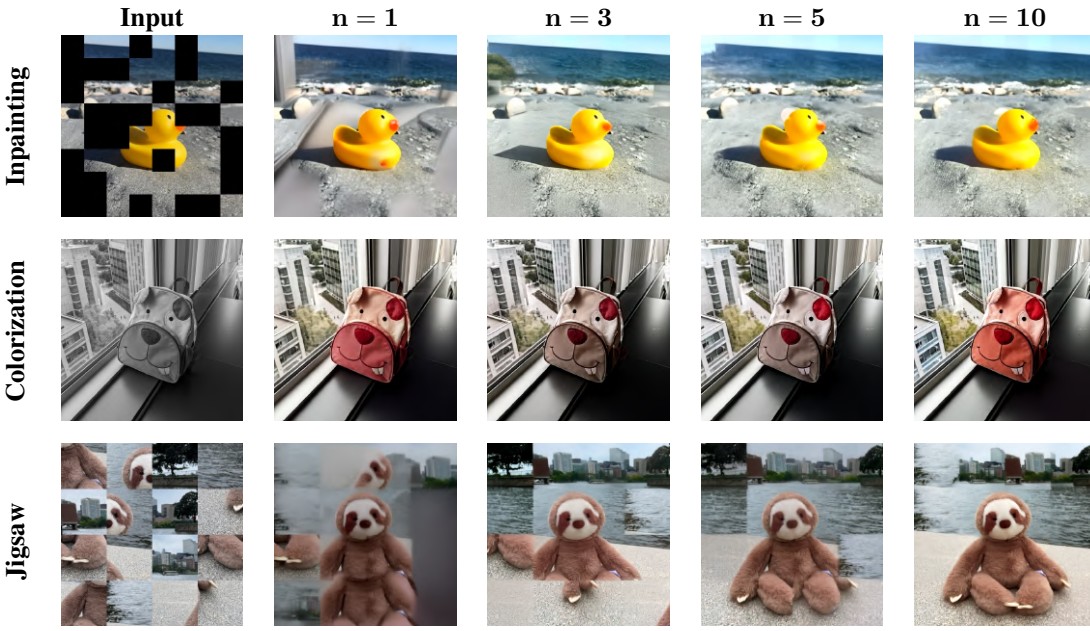

*Figure 32.* Qualitative results for different tasks (*Inpainting*, *Colorization*, *Jigsaw*) with different numbers of training examples.

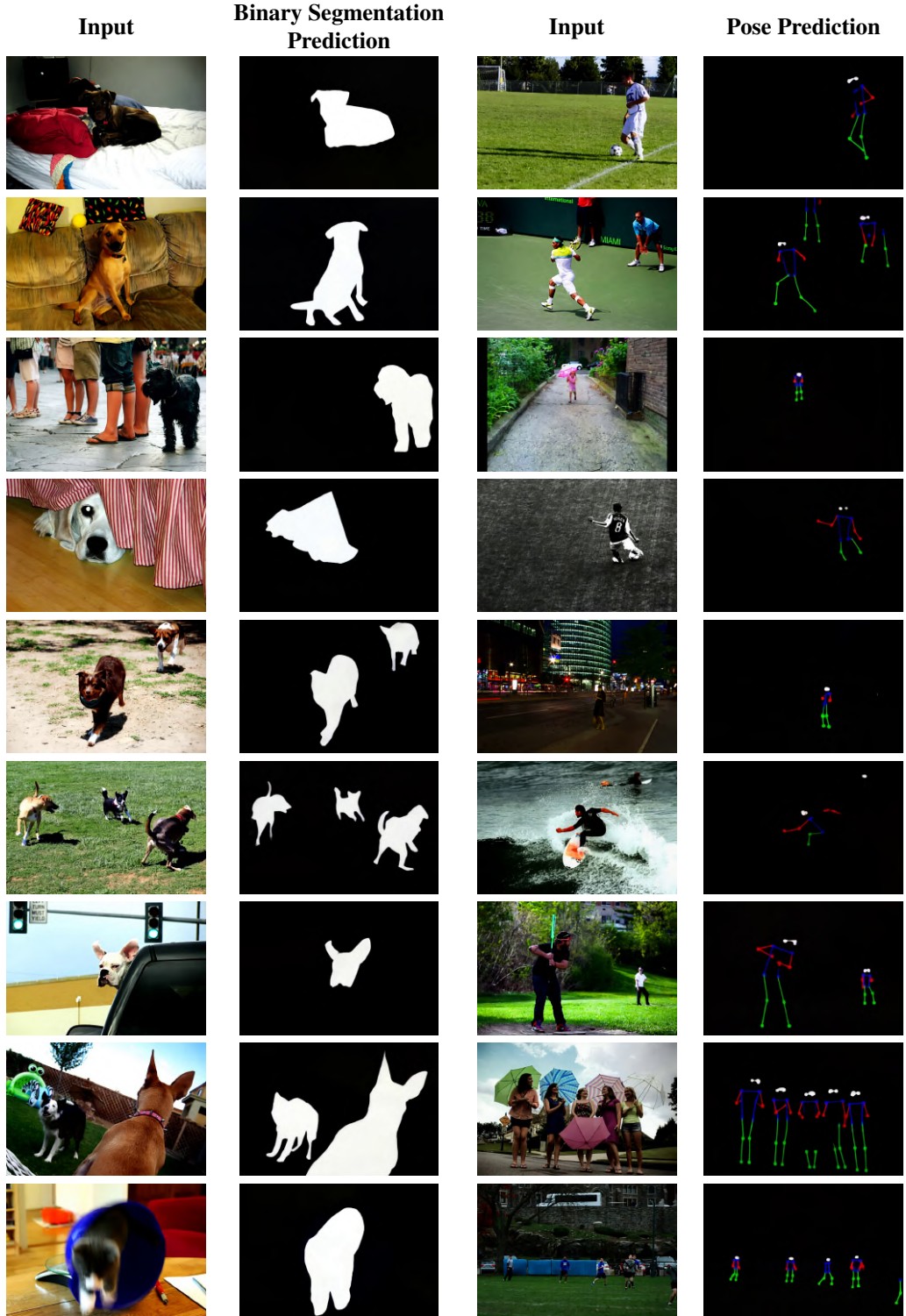

*Figure 33.* Predictions after finetuning with $n = 30$ samples for *Binary Segmentation* and *Pose*.

**Input**   **Depth Prediction**

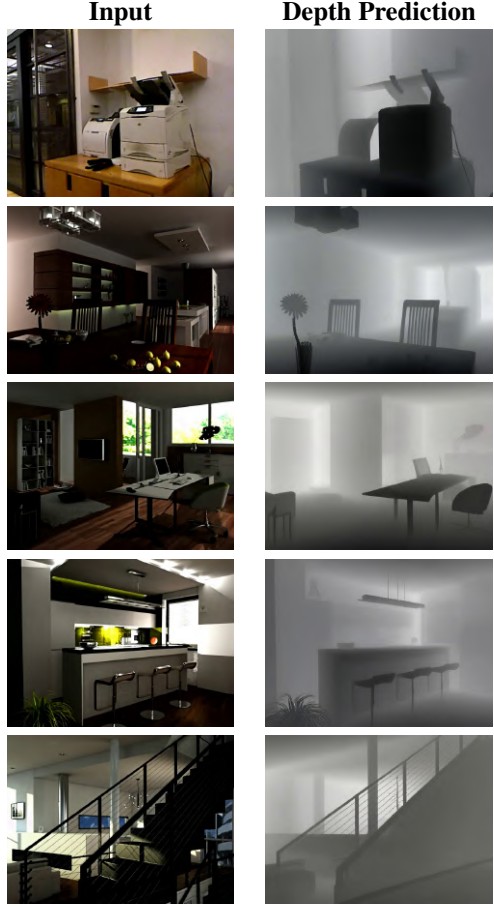

*Figure 34.* Predictions after finetuning with $n = 30$ samples for *Depth*.

**Input Image**   **Segmentation Prediction**

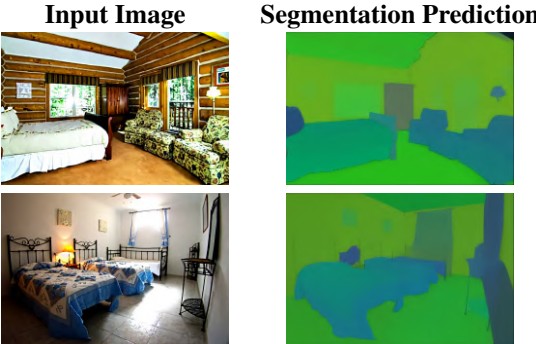

*Figure 35.* Examples from the *Image → Segmentation* in 1-shot setting for *Chamber*.

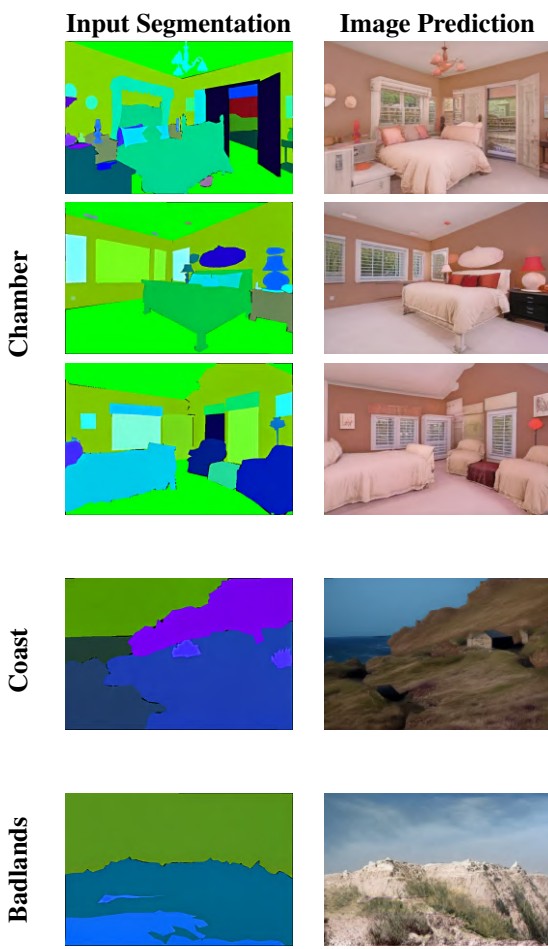

*Figure 36.* Examples from the *Segmentation → Image* task in the 1-shot setting. Each environment corresponds to a separate 1-shot training: for *Chamber* we train on one chamber and test on others, while for *Coast* and *Badlands* the same protocol applies within their category.

