# OpenReview forum: "Rethinking Visual Intelligence: Insights from Video Pretraining"
_ICML.cc/2026/Conference — ICML 2026 regular_

### Official Review · Reviewer_exfL · 2026-03-10

**Soundness:** 2
**Presentation:** 3
**Significance:** 3
**Originality:** 2
**Overall Recommendation:** 4
**Confidence:** 3

**Summary:**

This paper investigates visual reasoning in video diffusion models (VDM) and large language models (LLM). It proposes a systematic comparison of VDM and LLM on a various set of grid-based tasks (ARC-AGI, visual games, path-finding and cellular automata). VDM uses sequence of input/output images to process each reasoning task while LLM first converts the tasks into text-based format. Both paradigms (VDM, LLM) use the same lightweight adaptation protocol (LoRA) to adapt the base model to the specific tasks.

Across the various tasks, results tend to show that VDM are more sample-efficient and therefore that spatial priors are useful to solve those tasks.

**Compliance With Llm Reviewing Policy:**

Affirmed.

**Final Justification:**

Paper that investigates an important problem: demonstrating visual reasoning capabilities in VDMs. Paper did clarify some of concerns regarding generalization to other LLM model and ablation/design choices.

**Key Questions For Authors:**

See weaknesses.

**Limitations:**

Author adequately discuss the limitation of the approach.

**Strengths And Weaknesses:**

Strengths:
- It is a thought-provoking paper that investigates an important problem: demonstrating visual reasoning capabilities in VDMs.
- The authors propose a controlled evaluation setup that allows a 1:1 comparison between LLMs and VDMs.
- The authors perform a systematic ablation of training set size to investigate the sample efficiency of VDMs and LLMs on visual reasoning tasks.

Weaknesses:
- The main claim of the paper—that VDMs benefit from video pretraining for visual intelligence—is not fully supported by the empirical results. On ARC-AGI (Table 1), LLMs such as o1 or Sonnet achieve the best performance and outperform the CogVideoX VDM model. Then, in the rest of the paper, the authors only compare the VDM model to Qwen 3. It is unclear to me whether the conclusion would generalize to other models, and I would encourage the authors to report results on additional LLM baselines.

-The paper also lacks ablations regarding the main design choices. Why do you use discrete interpolation to generate the video sequence? How sensitive is the approach to the LoRA hyperparameters? How do you select the frame count and VDM architecture?

---

> ### Author Rebuttal · Authors · 2026-03-29
>
> We sincerely thank the reviewer for their time, feedback, and recognition of our paper as a thought provoking investigation into the visual reasoning capabilities of Video Diffusion Models. We appreciate the opportunity to clarify our experimental design and direct attention to several results that address the raised concerns.
>
> **Regarding the comparison with commercial Large Language Models**
>
> We respectfully point out that comparing a 5B parameter VDM to proprietary commercial models like OpenAI o1 preview or Claude 3.5 Sonnet is not a balanced assessment of capabilities. Those commercial models operate at a massively larger scale, estimated to be in the hundreds of billions of parameters (>100B). Our goal was to evaluate the inductive biases of the modalities themselves, which requires a controlled environment. When comparing models of similar capacity, the VDM demonstrates a significant advantage. As shown in Table 1, CogVideoX 1.5 5B achieves a 16.75 percent accuracy on ARC AGI, which is more than double the 8.00 percent achieved by the comparably sized Qwen3 4B Instruct.
>
> **Regarding generalization to other Large Language Models**
>
> Our conclusions are supported by comparisons with other language model families. We direct the reviewer to Figure 1 and Table 10, which explicitly include results for Llama 3.1 8B Instruct on the ConceptARC benchmark. The performance trends remain consistent across these models, affirming that our findings are not isolated to the Qwen family. We focused our deeper structured task analysis on Qwen3 primarily because it represented a state of the art open weight model at a parameter count comparable to the VDM, ensuring the fairest possible head to head evaluation.
>
> **Regarding ablations and design choices**
>
> Extensive ablation and hyperparameter optimization specifically tailored for the VDM would introduce a tuning bias, which would undermine the symmetric and fair comparison we aimed to establish with the language models.
>
> * Interpolation method: We selected discrete interpolation as the most straightforward setting to avoid introducing manual biases into the sequence. After, We have experimented with other interpolation methods, such as linear and quadratic, and found the results to be roughly similar.
> * LoRA sensitivity: The adaptation method is robust and largely insensitive to LoRA hyperparameters within a reasonable range.
> * Frame count and Architecture: We utilize the minimum number of frames, exactly 9, required to run the VAE meaningfully in our image to image setting. Regarding the architecture, we do not modify the internal structures of the models; our method relies on standard, frozen pretrained VDMs as described in their original foundational papers, adapting them solely through lightweight LoRA modules to isolate the impact of the pretraining.
>
> We hope this clarifies our methodological choices and highlights the comparisons already present in the manuscript.

---

> > ### Author Rebuttal · Reviewer_exfL · 2026-04-01
> >
> > Thank you for your rebuttal. I acknowledge that the paper provides comparison to Llama 3 family of models in addition to Qwen 3. However, I still think that the empirical findings depict a more nuanced picture than "VDMs benefit from video pretraining for visual intelligence". The data-efficient argument seems to hold for a certain class of models (below 5B) and for certain baselines (Qwen vs CogVideo), but it is unclear to me if this finding will generalize to large-scale models/different baselines.
> >
> > Overall, I find the paper to be an interesting read which asks important questions and think it could be a good fit for ICML.

---

> > > ### Author Response · Authors · 2026-04-02
> > >
> > > Thank you for your continued engagement. It is really encouraging to hear that you find it a good fit for ICML. We appreciate your perspective on the nuances of model scale and visual intelligence.
> > >
> > > While exploring exhaustive scales across all baselines is an extensive undertaking, we believe our current results on the ConceptARC benchmark (Table 10, Appendix) and Figure 1 offer compelling evidence that these benefits generalize to larger scales:
> > >
> > > **Evidence of Scaling and Generalization**
> > > We observe a consistent upward trend in performance as VDM scale increases, moving from LTX-2B (0.19 average accuracy) to CogVideoX1.5-5B (0.33) and Wan2.1-14B (0.41). In contrast, the language models we tested showed a noticeable plateau; e.g., doubling the scale from Qwen3-4B-Instruct to Qwen3-8B resulted in no improvement in average accuracy (both remaining at 0.24). This suggests that for visually grounded tasks, the spatiotemporal priors of a VDM provide a more effective foundation for scaling than simply increasing the parameter count of a text-based model. While these specific comparisons differ from our data-size ablations, they demonstrate superior adaptability in low-data settings (2-4 samples) across different architectures up to 14B.
> > >
> > > **Sample Efficiency and Future Scope**
> > > We agree that definitively proving this across every conceivable baseline and scale requires further validation. While the trajectory from 2B to 14B on ConceptARC suggests these spatiotemporal scaling benefits are robust, compute and time constraints during the rebuttal period don't allow for exhaustive ablations for the remaining task families.
> > >
> > > We acknowledge this as a limitation of our current empirical scope, but we hope these initial findings serve as a solid foundation for the community to further explore these larger-scale dynamics. Thank you again for helping us refine the impact and boundaries of our work.

---

### Official Review · Reviewer_1kML · 2026-03-10

**Soundness:** 2
**Presentation:** 3
**Significance:** 3
**Originality:** 3
**Overall Recommendation:** 5
**Confidence:** 4

**Summary:**

This paper investigates Video Diffusion Models (VDMs) as candidates for visual foundation models by reframing image-to-image tasks as transition video construction and leveraging spatiotemporal priors from video pretraining.

**Compliance With Llm Reviewing Policy:**

Affirmed.

**Final Justification:**

The authors give a reasonable clarification, I prefer to raise my scores.

Such a emperical study is meaningful for the exploration on the reasoning mechanisms behind modern large models. Moreover, I think it is necessary to validate the conclusions on more video-language deeply fused models, such as show-o2.

**Key Questions For Authors:**

1. Would VLM baselines narrow the gap from VDMs?

2. Do results transfer to real visual tasks, e.g., robotic manipulation?

**Limitations:**

Yes.

**Strengths And Weaknesses:**

Strengths:

1. Important and novel thinking perspective. The problem has further explored from architecture comparison to inductive bias comparison (spatiotemporal vs symbolic priors).

2. Rigorous controlled experimental design. The study ensures a fair comparison between VDMs and LLMs by using identical LoRA-based adaptation, task representation aligned to each model’s modality, and consistent training/evaluation splits across all benchmarks.

3. Diverse and comprehensive benchmark suite. The evaluation covers abstract reasoning, visual puzzles/games, spatial navigation, and cellular automata.

Weakness:

1. Inadequate quantitative analysis of efficiency tradeoffs. The study does not provide comparison of training/inference time, computational cost, or memory usage between VDMs and LLMs across tasks.

2. Underexplored hybrid model. The paper identifies clear complementary strengths between VDMs (2D spatial/dynamic reasoning) and LLMs (symbolic/linear sequence reasoning) but does not explore or evaluate hybrid models that integrate these two types of priors, e.g., Show-o2. Meanwhile, the discussion of VLMs and other similar works is insufficient, e.g, ChronoEdit.

3. Task complexity limitation. The evaluated tasks have relatively fixed complexity (e.g., fixed grid sizes for mazes, fixed prediction steps for cellular automata). The paper does not explore how VDMs and LLMs scale with increasing task complexity (e.g., larger grids, longer prediction horizons, more complex visual rules), so the scalability of the observed performance advantages is unknown.

---

> ### Author Rebuttal · Authors · 2026-03-29
>
> Thank you for the detailed review and for highlighting the novelty of our perspective, the rigor of our experimental design, and the comprehensiveness of our benchmark suite. We genuinely appreciate your constructive feedback and address your specific concerns below.
>
> **Efficiency Tradeoffs**
> Regarding the quantitative analysis of efficiency, our primary focus is on data efficiency during skill acquisition, which closely aligns with the definition of intelligence we adopted for this study. We do not claim to optimize for computational efficiency. However, we did include detailed wall clock training times in Appendix A, specifically in Tables 3 and 4. In our observations, the language models generally train faster but suffer from slower inference. Conversely, the video diffusion models take longer to train but are faster at inference.
>
> **Hybrid Models and Vision Language Models**
> You raised a great point about hybrid architectures and Vision Language Models. We dedicated Appendix E.1 to briefly exploring why current Vision Language Models struggle in these low data regimes. Our experiments showed that providing image inputs to a Vision Language Model offered no measurable improvement over text only baselines, and the model failed to extract meaningful structure from the visual inputs.
>
> Integrating approaches like ChronoEdit requires extensive fine tuning on both image to image and video tasks, which fundamentally contradicts our core focus on data efficiency and minimal supervision. Regarding multimodal architectures, our study serves as a controlled means to fairly compare the inductive biases of strictly visual models against strictly language models, therefore didn't include them in the primary head-to-head comparison.
>
> **Task Complexity**
> We respectfully push back on the perceived limitation regarding task complexity, as our benchmark suite does explicitly test scaling difficulty. For instance, we evaluate Langtons ant across increasing prediction horizons of 2, 3, 5, and 10 steps into the future. We also test spatial generalization by training models on small 13 by 13 mazes and evaluating them on larger 21 by 21 mazes. Furthermore, the ARC benchmark inherently contains visual rules of varying complexity. We found that pushing the complexity to more extreme levels caused the language models to fail completely and much earlier than the video models. To get a meaningful comparative signal in those extreme cases, we would need to increase the training data to unreasonable levels.
>
> **Transfer to Real Visual Tasks**
> Finally, you asked if these results might transfer to real visual tasks like robotic manipulation. We provide initial evidence supporting this in Appendix G, where we demonstrate the frameworks applicability to classical computer vision tasks such as depth prediction, segmentation, and human pose estimation using very few examples. While creating a systematic way to compare both model families on embodied robotics is beyond the scope of this current paper, we completely agree it is a highly promising avenue for future research. We are excited to see the community starting to explore this exact direction.
>
> Thank you again for your time and for helping us improve the clarity of our work.

---

> > ### Author Rebuttal · Reviewer_1kML · 2026-04-02
> >
> > Thanks for the authors' response, my main concerns have been resolved.

---

### Official Review · Reviewer_kAHX · 2026-03-14

**Soundness:** 2
**Presentation:** 2
**Significance:** 2
**Originality:** 2
**Overall Recommendation:** 4
**Confidence:** 3

**Summary:**

This paper plans to do research on the helpfulness of pretraining video generation models to downstream tasks. They give out some simple adaptation methods and do experiment on it. Strengths

The overall motivation is solid — the paper discusses an important question of what pre-training can actually bring.

The paper includes a substantial number of experiments that are easy to follow and well-presented.

Weaknesses

The abstract is overly vague; it does not clearly state what was done or what the main contributions are. It mostly repeats well-known statements such as “to support progress toward visual foundation models.”

Figure 1 is quite difficult to understand. The caption introduces new terminology without explanation, and the figure itself is not clearly interpreted.

It would be helpful to include a structural diagram illustrating how your adaptation mechanism works.

The insights appear limited — the main conclusion (that pre-training and joint training on related tasks can help) is already a widely held assumption. The paper should provide new or less obvious findings to stand out.

**Compliance With Llm Reviewing Policy:**

Affirmed.

**Final Justification:**

The authors present a reasonably clear motivation; however, the abstract and figure explanations are currently somewhat unclear. In the rebuttal, the authors committed to improving these descriptions, which addresses the main concerns. Accordingly, I have increased my score.

**Key Questions For Authors:**

refere to above.

**Limitations:**

The authors should provide more practical suggestions and conduct a deeper analysis.

**Strengths And Weaknesses:**

Strengths
1. The overall motivation is solid — the paper discusses an important question of what pre-training can actually bring.
2. The paper includes a substantial number of experiments that are easy to follow and well-presented.

Weaknesses
1. The abstract is vague; it does not clearly state what was done or what the main contributions are. It mostly repeats well-known statements such as “to support progress toward visual foundation models.”
2. Figure 1 is difficult to understand. The caption introduces new terminology without explanation, and the figure itself is not clearly interpreted.
3. It would be helpful to include a structural diagram illustrating how your adaptation mechanism works.
4. The insights appear limited — the main conclusion (that pre-training and joint training on related tasks can help) is already a widely held assumption. The paper should provide new or less obvious findings to stand out.

---

> ### Author Rebuttal · Authors · 2026-03-29
>
> We sincerely thank Reviewer kAHX for their time and feedback. We are glad you found the overall motivation solid and the experiments easy to follow and well presented. Below we address your specific concerns.
>
> **Abstract Clarity**
> We appreciate your perspective on the abstract. While we believe the current text effectively outlines our investigation into Video Diffusion Models, our controlled evaluation design, and the specific benchmarks tested, we understand the desire for more explicit takeaways. In the revised manuscript, we will update the abstract to highlight our specific quantitative findings and more directly state our contributions regarding spatial understanding.
>
> **Clarity of Figure 1**
> We apologize if Figure 1 was difficult to interpret. As noted, the terminology corresponds directly to the established ConceptARC benchmark. The figure is a standard radar plot comparing model accuracy across these specific competencies. As the plot illustrates, Video Diffusion Models demonstrate better accuracy across most skills evaluated. To ensure this is clear to all readers, we will expand the caption to explicitly explain the radar plot format and add a new section in the appendix detailing the ConceptARC benchmark categories.
>
> **Structural Diagram for Adaptation Mechanism**
> Thank you for this excellent suggestion. We originally aimed to make the adaptation clear through our textual explanation in section "3.2. Adapting Video Diffusion Models for I2I". However, we completely agree that a visual diagram would enhance comprehension. While we cannot provide figures during the rebuttal phase, we commit to adding a side by side architectural diagram in the final version to clearly illustrate our apple to apple adaptation framework.
>
> **Novelty of Insights**
> We respectfully disagree that our findings are merely a widely held assumption. While it is known that pretraining helps in general, it is far from obvious that training a model on natural video generation endows it with the ability to solve highly abstract, discrete reasoning puzzles like ARC AGI or cellular automata. These tasks are fundamentally different from natural video domains. Demonstrating that spatiotemporal pretraining translates directly into strong inductive biases for abstract visual problem solving is a novel and significant finding for the development of visual foundation models.

---

> > ### Author Rebuttal · Reviewer_kAHX · 2026-04-04
> >
> > Thank you for the authors’ response. I will increase my score accordingly.

---

> > > ### Author Response · Authors · 2026-04-05
> > >
> > > We are very glad to hear that your concerns have been resolved.
> > >
> > > We noticed that the overall score still reflects the initial rating. As you mentioned you would increase your score accordingly, we would be incredibly grateful if you could update the rating in the system whenever you have a moment.
> > >
> > > Thank you for your constructive feedback and support of our work.

---

### Decision · Program_Chairs · 2026-04-30

**Decision:**

Accept (regular)

**Comment:**

This article adapts LLMs vs. generative video models to be able to solve visual reasoning tasks like ARC-AGI. The article is a refreshing perspective on data efficiency / ease of adaptation of video models vs. LLMs, and all reviewers are in favor of acceptance. While some questions around the generalization of the findings remain at larger scales, these could be addressed by an honest limitations / caveats section, which the authors are encouraged to add for the final version.